 SciPost Phys. Lect.Notes 50 (2022)

# Effective theory for matter in non-perturbative cavity QED

**Juan Román-Roche and David Zueco**

Instituto de Nanociencia y Materiales de Aragón (INMA), CSIC-Universidad de Zaragoza,
Zaragoza 50009, Spain

## Abstract

Starting from a general material system of $N$ particles coupled to a cavity, we use a coherent-state path integral formulation to produce a *non-perturbative* effective theory for the material degrees of freedom. We tackle the effects of image charges, the $A^2$ term and a multimode arbitrary-geometry cavity. The resulting (non-local) action has the photonic degrees of freedom replaced by an effective position-dependent interaction between the particles. In the large-$N$ limit, we discuss how the theory can be cast into an effective Hamiltonian where the cavity induced interactions are made explicit. The theory is applicable, beyond cavity QED, to any system where bulk material is linearly coupled to a diagonalizable bosonic bath. We highlight the differences of the theory with other well-known methods and numerically study its finite-size scaling on the Dicke model. Finally, we showcase its descriptive power with three examples: photon condensation, the 2D free electron gas in a cavity and the modification of magnetic interactions between molecular spins; recovering, condensing and extending some recent results in the literature.

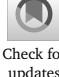

# 1   Introduction

The development of cavity QED is marked by the achievement of strong light-matter coupling [1]. This experimental milestone opened the possibility of entangling light and matter. Since then, experiments have advanced to be able to control and manipulate a few atoms and photons in a fully quantum way. Nowadays, there are many situations where strong coupling is achieved. Natural atoms or molecules as well as artificial systems such as quantum dots or superconducting circuits [2]. Some of these systems allow us to push light-matter coupling to new regimes. One of them is ultrastrong coupling, where the interaction strength is comparable to the bare energies of cavity photons and atomic transitions [3,4]. Here, correlations arise such that the ground state is already an entangled state with both virtual matter and photon excitations [5,6]. In the same spirit as the original motivation of cavity QED -to control atoms with quantum, rather than classical, light- a new field of research is emerging. It is known as cavity QED materials [7–9]. Here, instead of a few atoms, the modification of bulk material properties due to nonperturbative light-matter correlations is investigated. Thus, the focus is on the physics of a macroscopic number of particles coupled to the quantized electromagnetic field. Quantum light fluctuations have recently been shown to modify chemical reactivity [10, 11], excitonic transport [12–14], superconductivity [15–18] the ferroelectric phase in quantum paramagnetic materials [19–21] and ferromagnetism [22,23].

At the same time, the cavity-QED community has gained awareness of the critical role that certain ubiquitous approximations can have in the predicted behaviour of a system, often leading to wrong theoretical predictions that later fail to achieve experimental confirmation. This is especially important in the non-perturbative regime, which is precisely the regime where modifications of matter occur. Perhaps the most discussed example in the recent literature is equilibrium superradiance (photon condensation). This quantum phase transition was originally predicted for electric dipoles in a uniform electric field (Dicke model) [24–26], but it has since been shown to be just a consequence of the breaking of gauge invariance in theories missing the infamous $A^2$ term [27,28]. Equilibrium superradiance can be achieved, nevertheless, in a system of electric charges with spatially-varying electromagnetic (EM) fields [29–31] or in systems with magnetic dipoles [22], even when gauge invariance is restored, but it serves as an example of the profound consequences that an incomplete description of cavity QED can have. Several works have also pointed out that the two-level approximation is a source of mistakes if performed, without additional care, in the Coulomb gauge [32,33]. Finally, image

charges constitute a potentially relevant electrostatic contribution to the effective interactions between charges in a cavity [34,35]. Thus, a complete formulation of cavity QED must include them along the dynamical transverse fields of the cavity in the Coulomb gauge. This is crucial to later resolve the contributions of the two to the modification of matter properties.

Another important caveat in the study of hybrid systems in cavity QED is the fact that, when studying the interplay of light and matter in a strong coupling scenario, the separation between "light" and "matter" blurs. This occurs not only because of the well-understood creation of hybrid polaritonic excitations but, more fundamentally, because the definition of "light" and "matter" subsystems is gauge-dependent [36]. This observation is perfectly compatible with the gauge invariance of physical observables but it raises questions regarding our labelling of the underlying cause of certain phenomena [32,37]. When we say that an effective interaction is *light mediated*, is this a universal statement?, or can a change of gauge make it seem as though the interaction is *matter mediated*? Here, we avoid descending into this philosophical rabbit hole by defining the bare matter first. We start by establishing the Hamiltonian that describes the behaviour of a material system in the absence of a confined electromagnetic field, i.e. without a cavity. This is motivated by the fact that most material properties are well captured by such a description, such as superconductivity, ferromagnetism, electric and thermal conductivity, etc.. In this sense, there exist experimentally accessible observables that can be measured for the bare material system. Then, we study what happens when such a material is placed within a cavity. For that, we couple it to the quantized electromagnetic field in the Coulomb gauge for an arbitrary cavity geometry. The *experimentally relevant question* is then: if we measure the same experimentally accessible observables now, to what extent are they altered? If there is a change, we naturally attribute it to the effect of the cavity, even if in a microscopic description of the hybrid light-matter system it is impossible to universally distinguish between the two subsystems.

In this work we develop the theoretical analogue to this experimental procedure by deriving an effective theory of matter inside a cavity. Generally, an effective theory is found when a subsystem is systematically eliminated from the dynamical description of a larger system. Perturbative effective theories have widespread use in cavity QED. Notable examples are adiabatic elimination [38] or the dispersive theory [39–41]. However, for our purpose of studying strong light-matter correlations, it is necessary to consider non-perturbative techniques. A seminal example is the Feynman-Vernon influence functional within the path integral formalism, developed for quantum systems coupled to a continuum (open systems) [42–45]. Recently, coherent state path integrals have been used to define effective models in cavity QED [46–48]. Here, we build upon these works: we use a coherent-state path integral to define an *exact effective theory* for a general material gauge-invariantly coupled to a general cavity. This exact treatment yields a non-local effective action that becomes local in the large-$N$ limit. In this limit, the theory can be cast into an effective matter Hamiltonian where the effect of light vacuum fluctuations takes the explicit form of induced matter-matter interactions. The range and form of these interactions depends on the light-matter Hamiltonian and the geometry of the cavity. The theory is compared against those effective models commonly employed in the field. Unlike theories based on the adiabatic elimination of fast variables, our theory is valid in the full light-matter coupling regime. We also test its finite size scaling numerically via exact diagonalization. In doing that, we show that our theory matches the exact results already for $N \sim 10^2$. Finally, we test the power of our formalism by deriving recent results from the literature.

The remainder of the manuscript is organised as follows. In Section 2 we provide a graphical and non-technical summary of the main results of the paper. Section 3 is devoted to constructing a comprehensive model of matter inside an electromagnetic cavity. Then, Section 4 constitutes the main result of the paper: we trace out the cavity to derive the effective model. In Sec. 5 we compare the theory against other effective models and test its finite size

scaling. In Sec. 6 we study different material systems within a cavity to showcase the descriptive power of our effective theory. We close the paper with some conclusions. Technical details and concepts of minor importance are sent to the appendices.

## 2 Summary of main results

The main focus of this work is the theoretical description of a broad range of materials when these are coupled to an electromagnetic cavity. To avoid subtleties, we fix our starting hypothesis. We begin by asserting that a material is an extensive collection of typically a macroscopic number $N$ of particles that can be isolated (to a good approximation) and their behaviour

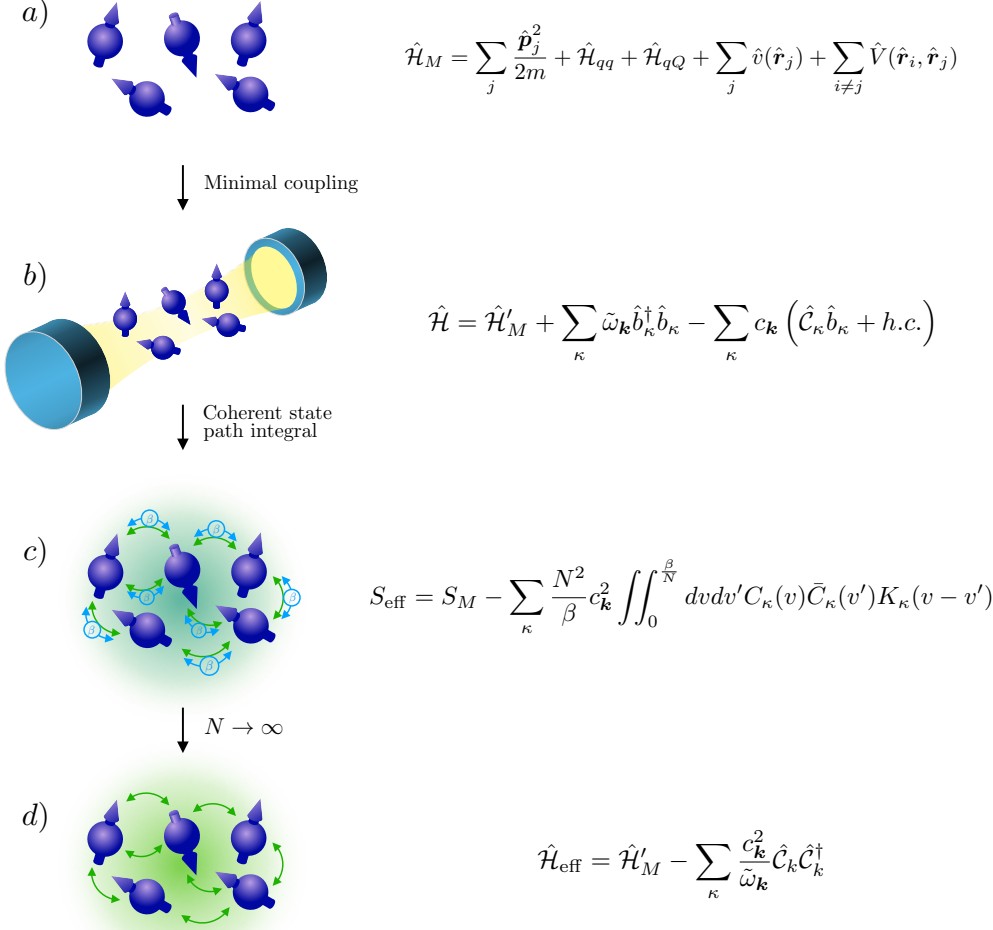

Figure 1: Schematic summary of the derivation of the effective theory. We begin by defining a general matter model in a). We accompany the illustration with an example Hamiltonian, but the theory extends to other models as well. The matter is then minimally coupled to an electromagnetic cavity in b). The corresponding Hamiltonian describes a broad range of models, which are encoded in the coupling constants $\{c_{\boldsymbol{k}}\}$ and operators $\{\hat{\mathcal{C}}_{\kappa}\}$, and in $\hat{\mathcal{H}}'_M$. In c), the light degrees of freedom are non-perturbatively eliminated with a coherent state path integral, yielding a non-local action with effective matter-matter couplings. The non-local contributions vanish in the large-$N$ limit and an exact effective Hamiltonian is obtained in d).

described by a Hamiltonian $\hat{\mathcal{H}}_M$. This is illustrated in Fig. 1(a). We then couple this material to a quantized electromagnetic cavity and our formalism begins. For the cavity, we focus on a complete description including the $A^2$ term, the Zeeman coupling and image charges from the cavity boundaries. We consider a multi-mode and non-uniform electromagnetic field and we do not specify any particular geometry, leaving the spatial dependence encoded in the mode functions of the EM field. We must disclaim that, in order to diagonalize the photonic Hamiltonian in the model of a gas of charged particles, we simplify certain mode-mode interactions arising from the $A^2$ term. This is a common occurrence in the literature and its adequacy is discussed in the text. We show that due to the fact that the light-matter Hamiltonian is quadratic in bosonic (light) operators, it can be brought to the linear form shown in Fig. 1(b), which corresponds to Eq. (17). This is a general Hamiltonian that can be used to describe other systems besides the ones that we treat in this paper. It describes the linear coupling of an unspecified extensive subsystem of interest to a diagonal(ized) bosonic ensemble. In our case, the role of the bosons is played by cavity photons, but the Hamiltonian could also be used to describe, e.g., phononic or magnonic ensembles.

Then, we use a coherent-state path integral formulation of the partition function of the system to trace out the photonic degrees of freedom (DOFs). In an *exact* treatment, we obtain an effective action (33) in which the light-matter coupling has been replaced by an cavity-field-mediated interaction between the constituents of the material subsystem. This interaction is generally non-local in imaginary time, as illustrated in Fig. 1(c). Then, we show that in the large-$N$ limit the action becomes local, moving from an effective action to an effective Hamiltonian (41) as shown in Fig. 1(d). In this effective Hamiltonian, the role of the cavity is made explicit in two ways: first, the bare coulombic interactions are modified by the image charges; second, an interpretable additional term introduces effective interactions mediated by the transverse fields of the cavity. In the process, we recover relations between photonic and material observables, such that information about the cavity is accessible even when the latter is traced out. The theory is tested on the Dicke model against other effective techniques, showing that it is not equivalent to perturbative techniques . Its finite size scaling is also numerically tested, showing that our theory becomes exact already for $N \sim 10^2$. To illustrate the applicability of our formalism, we finish the manuscript by applying it to three different material systems in a cavity. First, we revisit the problem of photon condensation. Within our effective model, we recover the no-go theorem for superradiance in the single-uniform-mode limit and we show how neglecting the $A^2$ term leads to a wrong prediction of the phase transition. Our result rules out phase transitions of any order in a single formulation and is valid at finite temperatures. This is a generalization of [28, 49]. Then, we obtain critical conditions for photon condensation with spatially-varying electromagnetic fields, recovering [31]. Then, we solve the 2D free electron gas in a cavity, recovering the results in [50]. The last system that we solve is an ensemble of molecular nanomagnets. These couple to the cavity solely by the Zeeman term, avoiding the $A^2$ problem, and have been shown to be a suitable candidate to showcase cavity-modified ferromagnetism as well as photon condensation [22]. We obtain the effective spin-spin interactions for an arbitrary magnetic field geometry.

## 3 Model

We begin by fixing the model under consideration. First, we consider a general class of material subsystems, composed of $N$ identical particles. Depending on the degree to which the particles are bounded to a particular position in space we can distinguish two different families of systems. A gas, where the particles are free to occupy any position of space, like an electron gas in a metallic lattice; and a system of emitters where each particle or group of particles is

bounded to a non-dynamical core, such as a system of neutral atoms or molecules where the motion of the nucleus is disregarded and the focus is placed on the electron(s) motion. We also discuss the limit case of completely fixed emitters such as magnetic spins. The two families differ slightly in the way they couple to EM fields. Nevertheless, the degree of parallelism in their analytical description is high, so we will only discuss the more complex case of the gas in the main text, and comment briefly on the derivation for a system of localized emitters, highlighting the differences with the former and reserving the detailed derivation for Appendix A.

## 3.1 A gas of charged particles

A gas of charged particles is generally described by a Hamiltonian of the form

$$\hat{\mathcal{H}}_M = \sum_j \frac{\hat{\boldsymbol{p}}_j^2}{2m} + \hat{\mathcal{H}}_{qq} + \hat{\mathcal{H}}_{qQ} + \sum_j \hat{v}(\hat{\boldsymbol{r}}_j) + \sum_{i \neq j} \hat{V}(\hat{\boldsymbol{r}}_i, \hat{\boldsymbol{r}}_j). \tag{1}$$

Here, $\hat{\boldsymbol{r}}_j$ and $\hat{\boldsymbol{p}}_j$ are the canonical position and momentum operators of the $j$-th particle, such that $[\hat{r}_{i,\alpha}, \hat{p}_{j,\beta}] = i\delta_{ij}\delta_{\alpha\beta}$. Note that $\hbar = 1$ throughout the text. We also include four potential terms. Coulombic interactions between charges, including self-interactions, are encapsulated in $\hat{\mathcal{H}}_{qq}$. $\hat{\mathcal{H}}_{qQ}$ contains possible interactions with non-dynamical charges, e.g. with an ionic lattice. The potentials $\hat{v}(\hat{\boldsymbol{r}}_j)$ and $\hat{V}(\hat{\boldsymbol{r}}_i, \hat{\boldsymbol{r}}_j)$ represent additional unspecified single-particle and two-particle potentials, respectively. This matter Hamiltonian can be summarized as $\hat{\mathcal{H}}_M = \hat{\mathcal{T}} + \hat{\mathcal{H}}_{qq} + \hat{\mathcal{H}}_{qQ} + \hat{\mathcal{V}}$. We are interested in the effects of coupling this material subsystem to the quantized electromagnetic field in a confined region of space, i.e. a cavity. To be consistent with the inclusion of Coulombic interactions, we will describe the coupling of the matter and light subsystems in the Coulomb gauge. This yields a Hamiltonian in minimal coupling format for the complete system

$$\hat{\mathcal{H}} = \sum_j \frac{(\hat{\boldsymbol{p}}_j - q\hat{\boldsymbol{A}}(\hat{\boldsymbol{r}}_j))^2}{2m} + \hat{\mathcal{H}}'_{qq} + \hat{\mathcal{H}}'_{qQ} + \hat{\mathcal{V}} + \hat{\mathcal{H}}_{\text{EM}} - \frac{g\mu_B}{2} \sum_j \hat{\boldsymbol{\sigma}}_j \hat{\boldsymbol{B}}(\hat{\boldsymbol{r}}_j). \tag{2}$$

The effect of the coupling is fourfold. Most trivially, we must account for the energy of the electromagnetic field, hence $\hat{\mathcal{H}}_{\text{EM}}$. Secondly, the canonical momentum now differs from the mechanical momentum, this is apparent in the kinetic term, which now includes the vector potential $\hat{\boldsymbol{A}}(\hat{\boldsymbol{r}}_j)$ at each particles' position. The third and perhaps more subtle effect is the modification of the Coulombic interactions between charges $\hat{\mathcal{H}}_{qq} \rightarrow \hat{\mathcal{H}}'_{qq}$, $\hat{\mathcal{H}}_{qQ} \rightarrow \hat{\mathcal{H}}'_{qQ}$, which now include direct as well as image-charge mediated interactions [34, 35]. The latter arise as an electrostatic effect of imposing the boundary conditions of a cavity on the Coulomb potential. Finally, we include the Zeeman coupling between the particles' spin and the magnetic field $\hat{\boldsymbol{B}}(\hat{\boldsymbol{r}}_j)$. Here $\hat{\boldsymbol{\sigma}}_j = (\hat{\sigma}_j^x, \hat{\sigma}_j^y, \hat{\sigma}_j^z)$ is the Pauli vector, whose components are the Pauli matrices, $\mu_B = \frac{e}{2m}$ is the Bohr magneton and $g$ the Landé factor. W.l.o.g. we are discussing spin $1/2$ particles. In fact, Eq. (2) is the many-particle generalization of the Pauli-Fiertz Hamiltonian for non-relativistic spin $1/2$ particles, bar the spin-orbit coupling.

In the Coulomb gauge, the redundancy in the description of the dynamical variables is eliminated by constraining the vector potential to be transverse, i.e. $\boldsymbol{\nabla} \cdot A = 0$. The quantized vector potential then reads

$$\hat{\boldsymbol{A}}(\hat{\boldsymbol{r}}_j) = \sum_\kappa A_{\boldsymbol{k}} \left( \hat{\boldsymbol{u}}_\kappa(\hat{\boldsymbol{r}}_j) \hat{a}_\kappa + \hat{\boldsymbol{u}}_\kappa^*(\hat{\boldsymbol{r}}_j) \hat{a}_\kappa^\dagger \right), \tag{3}$$

where $\kappa$ is a four-vector containing the polarization index $\sigma = 1, 2$ and the wavevector $\boldsymbol{k}$: $\kappa \equiv \{\boldsymbol{k}, \sigma\}$. Note that $A_{\boldsymbol{k}} = A_{-\boldsymbol{k}}$. We also introduce the bosonic annihilation and creation operators of the $\kappa$-th mode: $a_\kappa$, $a_\kappa^\dagger$, which obey the canonical commutation relations

$[a_\kappa, \hat{a}^\dagger_{\kappa'}] = \delta_{\kappa\kappa'}$. To maintain as much generality as possible, we have refrained from using a particular spatial dependence for the vector potential. For a specific model, the geometry of the cavity will determine the spatial quantization of the wavevector $\boldsymbol{k}$ and in turn, the functional form of the mode functions $u_\kappa(\boldsymbol{r})$ [51]. In any case, the following properties hold for the mode functions: $u_{-\boldsymbol{k},\sigma}(\boldsymbol{r}) = u^*_{\boldsymbol{k},\sigma}(\boldsymbol{r})$, $u_{\boldsymbol{k},\sigma}(\boldsymbol{r}) \cdot \boldsymbol{k} = 0$, $\int_V dV u^*_\kappa(\boldsymbol{r}) u_{\kappa'}(\boldsymbol{r}) = \delta_{\kappa\kappa'}$. Note that in Eq. (3) the mode functions have been promoted to mode operators, since they depend on the position operator of each particle (Cf. App. A). By definition, $\hat{\boldsymbol{B}} = \boldsymbol{\nabla} \times \hat{\boldsymbol{A}}$, so

$$\hat{\boldsymbol{B}}(\hat{\boldsymbol{r}}_j) = \sum_\kappa B_{\boldsymbol{k}} \left( \hat{\boldsymbol{u}}_{\perp,\kappa}(\hat{\boldsymbol{r}}_j)\hat{a}_\kappa + \hat{\boldsymbol{u}}^*_{\perp,\kappa}(\hat{\boldsymbol{r}}_j)\hat{a}^\dagger_\kappa \right), \tag{4}$$

where we have defined the transverse mode functions $\boldsymbol{u}_{\perp,\kappa}(\boldsymbol{r}_j) = \frac{1}{|\boldsymbol{k}|}\boldsymbol{\nabla} \times \boldsymbol{u}_\kappa(\boldsymbol{r}_j)$ and $B_{\boldsymbol{k}} = A_{\boldsymbol{k}}|\boldsymbol{k}| = A_{\boldsymbol{k}}\omega_{\boldsymbol{k}}/c$. Finally, we can express $\hat{\mathcal{H}}_{\text{EM}}$ as

$$\hat{\mathcal{H}}_{\text{EM}} = \sum_\kappa \omega_{\boldsymbol{k}}\hat{a}^\dagger_\kappa\hat{a}_\kappa. \tag{5}$$

To facilitate the analytical treatment of the Hamiltonian, it is convenient to expand the kinetic term and substitute in Eqs. (3) and (4).

$$\begin{aligned}
\hat{\mathcal{H}} = \hat{\mathcal{H}}'_M + \sum_\kappa \omega_{\boldsymbol{k}}\hat{a}^\dagger_\kappa\hat{a}_\kappa &- \sum_j\sum_\kappa \frac{q}{m}\hat{p}_j A_{\boldsymbol{k}}\left(\hat{\boldsymbol{u}}_\kappa(\hat{\boldsymbol{r}}_j)\hat{a}_\kappa + h.c.\right) \\
&+ \sum_j \frac{q^2}{2m}\sum_{\kappa,\kappa'} A_{\boldsymbol{k}}A_{\boldsymbol{k}'}\left(\hat{\boldsymbol{u}}_\kappa(\hat{\boldsymbol{r}}_j)\hat{a}_\kappa + h.c.\right)\left(\hat{\boldsymbol{u}}_{\kappa'}(\hat{\boldsymbol{r}}_j)\hat{a}_{\kappa'} + h.c.\right) \\
&- \frac{g\mu_B}{2}\sum_j\sum_{\boldsymbol{k}} \hat{\boldsymbol{\sigma}}_j B_{\boldsymbol{k}}\left(\hat{\boldsymbol{u}}_{\perp,\kappa}(\hat{\boldsymbol{r}}_j)\hat{a}_\kappa + h.c.\right).
\end{aligned} \tag{6}$$

Here, $\hat{\mathcal{H}}'_M$ is just $\hat{\mathcal{H}}_M$ but with $\hat{\mathcal{H}}_{qq}$ ($\hat{\mathcal{H}}_{qQ}$) replaced by $\hat{\mathcal{H}}'_{qq}$ ($\hat{\mathcal{H}}'_{qQ}$) To condense the notation, we define the operators

$$\hat{U}_{\kappa,\kappa'} = N^{-1}\sum_j \hat{\boldsymbol{u}}_\kappa(\hat{\boldsymbol{r}}_j)\hat{\boldsymbol{u}}_{\kappa'}(\hat{\boldsymbol{r}}_j), \tag{7}$$

$$\hat{\bar{U}}_{\kappa,\kappa'} = N^{-1}\sum_j \hat{\boldsymbol{u}}_\kappa(\hat{\boldsymbol{r}}_j)\hat{\boldsymbol{u}}^\dagger_{\kappa'}(\hat{\boldsymbol{r}}_j), \tag{8}$$

and the constants

$$\Delta_{\boldsymbol{k},\boldsymbol{k}'} = \frac{Nq^2 A_{\boldsymbol{k}}A_{\boldsymbol{k}'}}{2m}, \tag{9}$$

$$c^e_{\boldsymbol{k}} = qA_{\boldsymbol{k}}\sqrt{\frac{\omega_{\boldsymbol{k}}}{m}}, \tag{10}$$

$$c^m_{\boldsymbol{k}} = \frac{g\mu_B B_{\boldsymbol{k}}}{2}. \tag{11}$$

With these, we write the terms that depend quadratically on the bosonic operators as

$$\hat{\mathcal{H}}_{\text{ph}} = \sum_\kappa \omega_{\boldsymbol{k}}\hat{a}^\dagger_\kappa\hat{a}_\kappa + \sum_{\kappa,\kappa'} \Delta_{\boldsymbol{k},\boldsymbol{k}'}\left(\hat{U}_{\kappa,\kappa'}\hat{a}_\kappa\hat{a}_{\kappa'} + \hat{\bar{U}}_{\kappa,\kappa'}\hat{a}_\kappa\hat{a}^\dagger_{\kappa'} + h.c.\right). \tag{12}$$

Analytical tractability demands that we neglect mode-mode interactions (Cf. App. A) and assume $\hat{U}_{\boldsymbol{k},\sigma,-\boldsymbol{k},\sigma} = \hat{\bar{U}}_{\boldsymbol{k},\sigma,\boldsymbol{k},\sigma} = \mathbb{I}$, leaving only the momentum-conserving terms, and without spatial dependence

$$\hat{\mathcal{H}}_{\text{ph}} \approx \sum_{\boldsymbol{k},\sigma} \omega_{\boldsymbol{k}}\hat{a}^\dagger_{\boldsymbol{k},\sigma}\hat{a}_{\boldsymbol{k},\sigma} + \sum_\kappa \Delta_{\boldsymbol{k}}\left(\hat{a}_{\boldsymbol{k},\sigma}\hat{a}_{-\boldsymbol{k},\sigma} + \hat{a}_{\boldsymbol{k},\sigma}\hat{a}^\dagger_{\boldsymbol{k},\sigma} + h.c.\right), \tag{13}$$

where $\Delta_k \equiv \Delta_{k,k}$. The approximation is exact for a homogeneous gas. Furthermore, for gasses that have a second order phase transition to a non-homogeneous phase, the approximation is valid in the vicinity of the transition [31]. This suggests that the resulting model can be used to predict and locate transitions to non-homogeneous gas phases. It is expected that the effect of the approximation will be quantitative and will skew the calculation of observables in the non-homogeneous phase. $\hat{\mathcal{H}}_{\text{ph}}$ can now be diagonalized with a Bogoliubov transformation

$$\hat{a}^\dagger_{k,\sigma} = \cosh(\theta_k)\hat{b}^\dagger_{k,\sigma} - \sinh(\theta_k)\hat{b}_{-k,\sigma}. \tag{14}$$

One finds that $\cosh(\theta_k) = (\lambda_k + 1)/(2\sqrt{\lambda_k})$ and $\sinh(\theta_k) = (\lambda_k - 1)/(2\sqrt{\lambda_k})$ and $\lambda_k = \sqrt{1 + 4\Delta_k/\omega_k}$. As a result, the complete Hamiltonian can be written as

$$
\begin{aligned}
\hat{\mathcal{H}} = \hat{\mathcal{H}}'_M &+ \sum_\kappa \omega_k \lambda_k \hat{b}^\dagger_\kappa \hat{b}_\kappa - \sum_j \sum_\kappa c^e_k \lambda_k^{-1/2}\sqrt{\frac{1}{m\omega_k}}\hat{p}_j\left(\hat{u}_\kappa(\hat{r}_j)\hat{b}_\kappa + h.c.\right) \\
&- \sum_j \sum_k c^m_k \lambda_k^{-1/2}\hat{\sigma}_j\left(\hat{u}_{\perp,\kappa}(\hat{r}_j)\hat{b}_\kappa + h.c.\right).
\end{aligned}
\tag{15}
$$

Defining the coupling operator and coupling constant

$$c_k \hat{\mathcal{C}}_\kappa = \sum_j \lambda_k^{-1/2}\left(c^e_k\sqrt{\frac{1}{m\omega_k}}\hat{p}_j\hat{u}_\kappa(\hat{r}_j) + c^m_k\hat{\sigma}_j\hat{u}_{\perp,\kappa}(\hat{r}_j)\right), \tag{16}$$

we can finally write the Hamiltonian as

$$\hat{\mathcal{H}} = \hat{\mathcal{H}}'_M + \sum_\kappa \tilde{\omega}_k \hat{b}^\dagger_\kappa \hat{b}_\kappa - \sum_\kappa c_k\left(\hat{\mathcal{C}}_\kappa \hat{b}_\kappa + h.c.\right), \tag{17}$$

where $\tilde{\omega}_k = \omega_k \lambda_k$.

## 3.2 A system of localized emitters

The main difference between a gas and a system of localized emitters relies on the fact that in the former case the dynamical particles are free to move on the entire volume of the system, whereas in the latter their movement is bounded to a non-dynamical core. A limit case is given by systems in which no movement occurs, and instead the dynamics affect some other degree of freedom, this is the case of spins of fixed positions, such as the ones that compose a magnetic crystal. If the length scale of the bounded motion is much smaller than the wavelength of the EM field, one can consider that the EM field is constant and equal at any point in the trajectory of an individual particle $\hat{A}(\hat{r}_j + R_j) \approx \hat{A}(R_j)$: where $R_j$ is the position of the corresponding non-dynamical core. This is the long-wavelength approximation. Assuming that we can work within the long-wavelength limit, the EM field operators $\hat{A}$, $\hat{B}$ act only on the Hilbert space of the photons, i.e. they do not contain position operators acting on the Hilbert space of the matter. Instead, they are parametrized by the positions of the non-dynamical cores. This subtle difference implies that, unlike for the gas, the terms in the Hamiltonian quadratic in bosonic operators can be exactly diagonalized by a Bogoliubov transformation. The final Hamiltonian (17) (See Fig. 1(b)) has the same functional form in both cases, the differences are encoded in the coupling constants $\{c_k\}$ and coupling operators $\{\hat{\mathcal{C}}_\kappa\}$, and of course in $\hat{\mathcal{H}}_M$. For a detailed derivation on the case of localized emitters see App. A.

Before advancing, we want to emphasize that Hamiltonian (17) or the equivalent (99) represents a general light-matter Hamiltonian. In this Section (Appendix A), we arrived to it by considering the particular case of a gauge-invariant description of a gas of charged particles

(an ensemble of localized emitters) coupled to the multi-mode electromagnetic field of a cavity, but this approach is not unique. For instance, the Zeeman coupling to the magnetic field is often neglected and the description of the EM field is simplified by making the single-mode approximation. Neglecting the $A^2$ term is also a ubiquitous approximation. The important fact is that all these variations arise as particularizations of Hamiltonian (17). More over, systems of magnetic molecules where the orbital degrees of freedom can be disregarded, which have received attention for the possibility of hosting the superradiant phase transition [22], can also be described by this Hamiltonian. In this case, the matter Hamiltonian $\hat{\mathcal{H}}'_M$ and the coupling operator $\hat{\mathcal{C}}_{\kappa}$ would exclusively contain spin degrees of freedom. In these examples, we simplify the picture by considering only cavity fields, but the matter DOFs can be coupled to classical external fields as well. For instance, it is common in systems of magnetic molecules to induce Zeeman level splitting with an external magnetic field perpendicular to the microwave magnetic field of the cavity. The inclusion of a classical field is trivial and indeed will be considered in Sec. 6 when we cover magnetic cavity QED.

We have discussed some relevant examples for cavity QED and quantum optics, but in general, any matter system which is linearly coupled to light and is quadratic in the photonic operators can have its Hamiltonian brought to Eq. (17). In essence, the formalism presented hereinafter is just a recipe to systematically eliminate bosonic degrees of freedom in a hybrid system. Furthermore, we show that in the large-$N$ limit, which is typically the relevant one in cavity QED materials, the effective action becomes local and the effective description for the matter is exactly Hamiltonian. Thus eliminating the need for mean field or perturbative approximations.

# 4 Deriving the effective matter model

Our goal in this section is to take a Hamiltonian of the form of Eq. (17) and trace out the photonic degrees of freedom. To do so, we will use a coherent path integral approach to define an effective matter action. One can foresee that the light-matter coupling present in (17) will translate into an effective matter-matter coupling that will appear in the effective action. Then, we obtain an effective Hamiltonian from the effective action in the, $N \to \infty$, thermodynamic limit.

## 4.1 Effective action: Euclidean path integral formulation

Let us recall that the partition function of a single-mode bosonic system described by a Hamiltonian $\hat{\mathcal{H}}$ can be written in Euclidean path integral form as

$$Z = \int \mathcal{D}(z, \bar{z}) e^{-S}, \tag{18}$$

with Euclidean action

$$S = \int_0^\beta (\bar{z} \partial_u z + H(z, \bar{z})) \, du, \tag{19}$$

where $z, \bar{z}$ are complex-valued continous fields satisfying the boundary condition $\bar{z}(0) = \bar{z}(\beta)$, $z(0) = z(\beta)$ and $\mathcal{D}(z, \bar{z}) = \lim_{N \to \infty} \prod_{n=1}^{N} d(z^n, \bar{z}^n)$, with $d(z^n, \bar{z}^n) = d \operatorname{Re} z^n d \operatorname{Im} z^n$ [52, chapter 4]. Here $H(z, \bar{z})$ is obtained by replacing $a$ by $z$ and $a^\dagger$ by $\bar{z}$ in the normal-ordered form of $\hat{\mathcal{H}}$, where $a(a^\dagger)$ are bosonic annihilation (creation) operators. This result is readily generalized to the multimode case

$$Z = \prod_{\kappa} \int \mathcal{D}(z_\kappa, \bar{z}_\kappa) e^{-S_\kappa}, \tag{20}$$

with

$$S_\kappa = \int_0^\beta (\bar{z}_\kappa \partial_u z_\kappa + H_\kappa(z_\kappa, \bar{z}_\kappa))\, du. \tag{21}$$

We now focus our attention on a hybrid light-matter system such as the one described by Hamiltonian (17). With $|\mathbf{z}\rangle = \bigotimes_\kappa |z_\kappa\rangle$, its partition function is given by

$$Z = \mathrm{Tr}_M \left( \int \prod_\kappa \frac{d^2 z_\kappa}{\pi} \langle \mathbf{z}| e^{-\beta \hat{\mathcal{H}}} |\mathbf{z}\rangle \right). \tag{22}$$

We assume that one could formulate a coherent-path-integral partition function for the full system, so we assign an action to the matter Hamiltonian $\hat{\mathcal{H}}'_M \to S_M$ and a field to the coupling operator $\hat{\mathcal{C}}_k \to C_k$, $\hat{\mathcal{C}}_k^\dagger \to \bar{C}_k$. Knowledge of the precise dependence of $S_M$, $C_k$ and $\bar{C}_k$ on material coherent fields is not required for what follows. With these definitions, the total action can be written as $S = S_M + \sum_\kappa S_\kappa$, with

$$S_\kappa = \int_0^\beta \left( \bar{z}_\kappa \partial_u z_\kappa + \tilde{\omega}_{\mathbf{k}} \bar{z}_\kappa z_\kappa - c_{\mathbf{k}} \left( C_\kappa z_\kappa + \bar{C}_\kappa \bar{z}_\kappa \right) \right) du. \tag{23}$$

For such a system, we can define an effective matter action by tracing only over the light degrees of freedom

$$Z = \mathrm{Tr}_M \left( e^{-S_M} \prod_\kappa \int \mathcal{D}(z_\kappa, \bar{z}_\kappa) e^{-S_\kappa} \right) =: Z_0 \, \mathrm{Tr}_M \left( e^{-S_{\mathrm{eff}}} \right). \tag{24}$$

At this point, $Z_0$ is just a constant prefactor. It should be noted that $S_{\mathrm{eff}}$ is useful because it makes explicit the influence of the cavity on the matter DOFs. In particular, the type of interactions that emerge. In addition, as we show in Sec. 4.3, no information is lost because light observables can be written in terms of matter observables, which can be calculated from $S_{\mathrm{eff}}$.

We are thus interested in computing

$$\prod_\kappa \int \mathcal{D}(z_\kappa, \bar{z}_\kappa) e^{-S_\kappa}. \tag{25}$$

Since the trajectories are periodic $z(0) = z(\beta)$, $\bar{z}(0) = \bar{z}(\beta)$, we can expand the fields as a Fourier series

$$z_\kappa(u) = \frac{1}{\sqrt{\beta}} \sum_n z_\kappa(\omega_n) e^{i\omega_n u}, \tag{26}$$

$$\bar{z}_\kappa(u) = \frac{1}{\sqrt{\beta}} \sum_n \bar{z}_\kappa(\omega_n) e^{-i\omega_n u}, \tag{27}$$

with the Matsubara frequencies $\omega_n = \frac{2\pi n}{\beta}$. Inserting Eqs. (26), (27) in the action (23) yields

$$S_\kappa = \sum_n \left( (i\omega_n + \tilde{\omega}_{\mathbf{k}}) \bar{z}_\kappa(\omega_n) z_\kappa(\omega_n) - c_{\mathbf{k}} z_\kappa(\omega_n) C_\kappa(\omega_n) - c_{\mathbf{k}} \bar{z}_\kappa(\omega_n) \bar{C}_\kappa(\omega_n) \right), \tag{28}$$

where

$$C_\kappa(\omega_n) = \frac{1}{\sqrt{\beta}} \int_0^\beta du\, C_\kappa(u) e^{i\omega_n u}, \tag{29}$$

$$\bar{C}_\kappa(\omega_n) = \frac{1}{\sqrt{\beta}} \int_0^\beta du\, \bar{C}_\kappa(u) e^{-i\omega_n u}. \tag{30}$$

The Jacobian of the Fourier transformation is unity, so the functional differential over temperature trajectories in Eq. (25) is simply replaced with the differential over frequency trajectories. The result is simply a collection of $n$-dimensional Gaussian integrals for each mode $\kappa$, yielding

$$\prod_\kappa \int \mathcal{D}(z_\kappa, \bar{z}_\kappa) e^{-S_\kappa} = \prod_\kappa Z_\kappa \exp\left[ \frac{c_k^2}{\beta \tilde{\omega}_k} \iint_0^\beta du du' C_\kappa(u) \bar{C}_\kappa(u') K_\kappa(u - u') \right], \qquad (31)$$

with the kernel

$$K_\kappa(u - u') = \sum_n \frac{\tilde{\omega}_k}{i\omega_n + \tilde{\omega}_k} e^{i\omega_n(u - u')}. \qquad (32)$$

At this point, the prefactor $Z_0$ defined in Eq. (24) acquires meaning: $Z_0 = \prod_\kappa Z_\kappa$, where $Z_\kappa = [1 - \exp(-\beta \tilde{\omega}_k)]^{-1}$ is the partition function of a Harmonic oscillator of frequency $\tilde{\omega}_k$. We can now put together the effective action as defined in Eq. (24)

$$S_{\text{eff}} = S_M - \sum_\kappa \frac{c_k^2}{\beta \tilde{\omega}_k} \iint_0^\beta du du' C_\kappa(u) \bar{C}_\kappa(u') K_\kappa(u - u'). \qquad (33)$$

After tracing out the light degrees of freedom, the light-matter interaction appears translated into a temperature-non-local interaction between matter degrees of freedom. The elimination of the cavity allows us to study the modification of material properties from the point of view of the matter subsystem. However, the non locality of the effective interaction implies that Eq. 33 does not have straightforward analytical applicability. In particular, we cannot recover an equivalent Hamiltonian. That is, we cannot undo our agnostic substitutions $\hat{\mathcal{H}}'_M \to S_M$, $\hat{\mathcal{C}}_k \to C_k$ and $\hat{\mathcal{C}}_k^\dagger \to \bar{C}_k$, to revert the description of the matter subsystem from an action-based one to a Hamiltonian-based one. This is because not all fields depend on the same temperature in this non-local action. This should not come as a surprise, since in classical statistical mechanics when integrating a subpart of the whole system the magnitudes that involve the remaining DOFs can be calculated from a temperature-dependent effective Hamiltonian [53]. Equivalently, in bipartite quantum systems, the (temperature dependent) Hamiltonian of mean force describes the equilibrium properties of a partition [54, 55]. On the other hand, (33) is formally analogous to the effective action after tracing out the environment DOFs in the path integral formulation of open quantum systems [44].

What is less intuitive is what we show in the next subsection, namely that the kernel becomes local in the thermodynamic limit $N \to \infty$, recovering a time-local effective interaction with a clear Hamiltonian equivalent.

## 4.2 Obtaining an effective matter Hamiltonian in the large-$N$ limit

The effective action (33) only depends on $N$ through the scaling of the involved fields, which they inherit from the corresponding operators. We assume that the light-matter Hamiltonian (17) is not ill-defined in the thermodynamic limit, i.e. that all its terms scale as $N$. Otherwise some of them would become negligible for $N \to \infty$, rendering the system exactly solvable and, thus, of little analytical interest. It is also important to note that in order to have a macroscopically well-defined thermodynamic limit the density of particles in the material $\rho = N/V$ must be finite, and thus the volume $V$ must scale like the number of particles $N$, i.e. $V \to \infty$ in the thermodynamic limit. This implies that $b$, $b^\dagger$ scale as $\sqrt{N}$ and $c_k$ scales as $1/\sqrt{N}$, with both $\hat{\mathcal{H}}'_M$ and $\hat{\mathcal{C}}_\kappa$ scaling as $N$.

To understand how a large $N$ affects the effective action it is best to exploit the scaling properties of the photonic operators before they are traced out. This is done following the reasoning of Wang and Hioe [25]. Because the photonic operators scale as $\sqrt{N}$, one can define

rescaled operators $b/\sqrt{N}$, $b^\dagger/\sqrt{N}$ whose $N \to \infty$ limit is well-defined and whose commutator scales as $1/N$, and thus vanishes in the large-$N$ limit. This allows one to write

$$\int \prod_\kappa \frac{d^2 z_\kappa}{\pi} \langle z | e^{-\beta \hat{\mathcal{H}}} | z \rangle = \int \prod_\kappa \frac{d^2 z_\kappa}{\pi} e^{-\beta \langle z | \hat{\mathcal{H}} | z \rangle}. \tag{34}$$

Defining

$$\tilde{Z} = \text{Tr}_M \left( \int \prod_\kappa \frac{d^2 z_\kappa}{\pi} e^{-\beta \langle z | \hat{\mathcal{H}} | z \rangle} \right), \tag{35}$$

with $|z\rangle = \bigotimes_\kappa |z_\kappa\rangle$ and

$$\langle z | \hat{\mathcal{H}} | z \rangle = \hat{\mathcal{H}}'_M + \sum_\kappa \tilde{\omega}_k |z_\kappa|^2 + \sum_\kappa c_k \left( \hat{\mathcal{C}}_\kappa z_\kappa + h.c. \right), \tag{36}$$

we note that taking the trace over matter of the lhs and rhs of Eq. (34) yields the equality $Z = \tilde{Z}$ (Cf. (22)). The same equality can be obtained from the bounds derived by Hepp and Lieb in their original study of equilibrium superradiance [26]

$$\tilde{Z} \leq Z \leq e^{\beta \sum_\kappa \omega_k} \tilde{Z}. \tag{37}$$

In the thermodynamic limit $N \to \infty$ and provided the number of modes does not scale with $N$, the correction term $\exp\left[\beta \sum_\kappa \omega_k\right]$ in the upper bound becomes negligible and the partition function is given exactly by $Z = \tilde{Z}$. With this equality at hand, we backtrack to Eqs. (23) and (24) and note that if one assumes constant photon fields, the path integral becomes a Gaussian integral over the initial states and Eq. (24) becomes precisely $\tilde{Z}$. Thus, in the path integral formulation, the large-$N$ limit implies constant paths for the photons. This resembles the classical limit in the coherent state path integral formulation of fields [56, chapter 2]. We may enforce constant fields at any point in the derivation of the effective theory. In particular, we may fulfill our initial desire of finding the large-$N$ limit of the final effective action (33). To do so, it is important to realize that the constant field condition equates to truncating the Matsubara summation to the $n = 0$ mode or, equivalently, to setting $\omega_n = 0$ for all $n$. This realization will allow us to enforce the consequences of the large-$N$ limit on the non-local kernel (32). Before doing so, it is convenient to do some manipulations

$$K_\kappa(\tau) = \sum_n \frac{\tilde{\omega}_k^2 e^{i\omega_n \tau}}{\omega_n^2 + \tilde{\omega}_k^2} - \sum_n \frac{i\omega_n \tilde{\omega}_k e^{i\omega_n \tau}}{\omega_n^2 + \tilde{\omega}_k^2} = \bar{K}_\kappa(\tau) - \frac{1}{\tilde{\omega}_k} \partial_\tau \bar{K}_\kappa(\tau). \tag{38}$$

The first term can be split into a collection of delta functions and another kernel

$$\bar{K}_\kappa(\tau) = \sum_n e^{i\omega_n \tau} - \sum_n \frac{\omega_n^2 e^{i\omega_n \tau}}{\omega_n^2 + \tilde{\omega}_k^2} = \beta \sum_n \delta(\tau - \beta n) - \bar{\bar{K}}_\kappa(\tau). \tag{39}$$

The delta functions provide the local action that we seek when integrating over the interval $[0, \beta]$. Crucially, $\bar{\bar{K}}_\kappa(\tau) = 0$ in the large-$N$ limit. To see this, recall that the large-$N$ limit equates to restricting to constant photonic fields, which in turn equates to setting $\omega_n = 0$ for all $n$. Returning to the second term in (38), because it is odd in $\tau$, it can be seen that it does not contribute to the action when the matter coupling operators are Hermitian and thus $C_\kappa = \bar{C}_\kappa$. In a more general case, it can contribute. Regardless, in the large-$N$ limit, it vanishes for the same reason as $\bar{\bar{K}}_\kappa$ and the action is purely local.

So in the thermodynamic limit the effective action becomes local in time

$$S_{\text{eff}} = S_M - \sum_\kappa N \frac{c_k^2}{\tilde{\omega}_k} \int_0^{\frac{\beta}{N}} dv\, C_\kappa(v) \bar{C}_\kappa(v) = S_M - \sum_\kappa \frac{c_k^2}{\tilde{\omega}_k} \int_0^\beta du\, C_\kappa(u) \bar{C}_\kappa(u). \tag{40}$$

Accordingly, we can recover the corresponding effective Hamiltonian

$$\hat{\mathcal{H}}_{\text{eff}} = \hat{\mathcal{H}}_M' - \sum_{\kappa} \frac{c_{\boldsymbol{k}}^2}{\tilde{\omega}_{\boldsymbol{k}}} \hat{\mathcal{C}}_k \hat{\mathcal{C}}_k^\dagger, \tag{41}$$

such that $Z = Z_0 \operatorname{Tr}_M \left[ \exp\left(-\beta \hat{\mathcal{H}}_{\text{eff}}\right) \right]$. An alternative derivation of the effective Hamiltonian can be found in App. B. Note also that what we show here is just a particular way of deriving the effective Hamiltonian, but constant photon paths can be enforced at any point along the derivation of the effective action, e.g. at the level of the kernel but before extracting the Dirac deltas. In those cases –the calculation may involve the use of the Hubbard-Stratonovich transformation– one always ends up with a Gaussian integral over photonic DOFs akin to the one discussed in App. B which also leads to the same effective Hamiltonian.

Equation (41) shows that in the large-$N$ limit the effect of the cavity can be described via an effective Hamiltonian in which the induced interactions are explicit (second term on the r.h.s. of Eq. (41)). Notice that sometimes similar effective interactions are argued by invoking approximations, such as perturbation or mean field theory. Here we show that they are exact in the thermodynamic limit for the matter. Besides, it is a rather general result. We emphasize that the cavity-matter coupling has been discussed under general grounds in Section 3, and Eq. (41) follows from it.

## 4.3 Relation between matter and light observables

Our effective theory puts emphasis on the behaviour of the matter subsystem, to the point where the cavity is eliminated from the Hamiltonian. However, for a complete characterization of the full system one must be able to compute photonic fields, $\langle b_\kappa \rangle$, $\langle b_\kappa^\dagger \rangle$, and the photon number which determine the intensity of the electromagnetic fields in the cavity. We reconcile this neccesity with our formalism by noting that in the process of tracing out the photonic degress of freedom we can extract a relationship between the expectation values of matter and light observables. To illustrate the idea, let us work out the case of computing $\langle b_{\kappa'} + b_{\kappa'}^\dagger \rangle$. We extend our Hamiltonian to the form $\hat{\mathcal{H}}(\lambda) = \hat{\mathcal{H}} - \lambda (b_{\kappa'} + b_{\kappa'}^\dagger)$, such that we can write, as is customary in statistical mechanics, $\beta \langle b_{\kappa'} + b_{\kappa'}^\dagger \rangle = \lim_{\lambda \to 0} \partial_\lambda \ln Z(\lambda)$, with $Z(\lambda) = \operatorname{Tr}\left(\exp\left[-\beta \hat{\mathcal{H}}(\lambda)\right]\right)$. At the same time, we can rewrite $\hat{\mathcal{H}}(\lambda)$ as

$$\hat{\mathcal{H}}(\lambda) = \hat{\mathcal{H}}_M' + \sum_{\kappa} \tilde{\omega}_{\boldsymbol{k}} \hat{b}_\kappa^\dagger \hat{b}_\kappa - \sum_{\kappa} c_{\boldsymbol{k}} \left( \hat{\mathcal{C}}_\kappa(\lambda) \hat{b}_\kappa + h.c. \right), \tag{42}$$

where $\hat{\mathcal{C}}_\kappa(\lambda) = \hat{\mathcal{C}}_\kappa - \delta_{\kappa,\kappa'} \lambda / c_{\boldsymbol{k}}$. Following our theory, the corresponding effective Hamiltonian is given by

$$\hat{\mathcal{H}}_{\text{eff}}(\lambda) = \hat{\mathcal{H}}_M' - \sum_{\kappa} \frac{c_{\boldsymbol{k}}^2}{\tilde{\omega}_{\boldsymbol{k}}} \hat{\mathcal{C}}_\kappa(\lambda) \hat{\mathcal{C}}_\kappa^\dagger(\lambda). \tag{43}$$

Then, following our definition of the effective action (41), we can express the partition function as $Z(\lambda) = Z_0 \operatorname{Tr}_M \left( \exp\left[-\beta \hat{\mathcal{H}}_{\text{eff}}(\lambda)\right] \right)$, such that

$$\left\langle b_{\kappa'} + b_{\kappa'}^\dagger \right\rangle = \lim_{\lambda \to 0} Z(\lambda)^{-1} Z_0 \operatorname{Tr}_M \left[ -\frac{c_{\boldsymbol{k}'}}{\tilde{\omega}_{\boldsymbol{k}'}} \left( \hat{\mathcal{C}}_{\kappa'}(\lambda) + \hat{\mathcal{C}}_{\kappa'}^\dagger(\lambda) \right) e^{-\beta \hat{\mathcal{H}}_{\text{eff}}(\lambda)} \right] = -\frac{c_{\boldsymbol{k}'}}{\tilde{\omega}_{\boldsymbol{k}'}} \left\langle \hat{\mathcal{C}}_{\kappa'} + \hat{\mathcal{C}}_{\kappa'}^\dagger \right\rangle_M. \tag{44}$$

Proceeding in similar fashion, we can compute other observables such as

$$\left\langle b_{\kappa'} - b_{\kappa'}^\dagger \right\rangle = \frac{c_{\boldsymbol{k}'}}{\tilde{\omega}_{\boldsymbol{k}'}} \left\langle \hat{\mathcal{C}}_{\kappa'} - \hat{\mathcal{C}}_{\kappa'}^\dagger \right\rangle_M. \tag{45}$$

Another observable of interest is the photon number of the $\kappa$-th mode

$$\left\langle b_{\kappa'}^\dagger b_{\kappa'} \right\rangle = -\frac{1}{\beta} \partial_{\tilde{\omega}_{\boldsymbol{k}'}} \ln Z = n_B(\beta, \tilde{\omega}_{\boldsymbol{k}'}) + \left( \frac{c_{\boldsymbol{k}'}}{\tilde{\omega}_{\boldsymbol{k}'}} \right)^2 \left\langle \hat{\mathcal{C}}_{\kappa'} \hat{\mathcal{C}}_{\kappa'}^\dagger \right\rangle_M, \tag{46}$$

where $n_B$ is the bosonic occupation number of a free Harmonic oscillator

$$n_B(\beta, \tilde{\omega}_{k'}) = \frac{1}{e^{\beta \tilde{\omega}_{k'}} - 1}. \tag{47}$$

We find that the photon number has two sources at finite temperature, one corresponding to the thermal fluctuations of the collection of free Harmonic oscillators that constitute the cavity, and another arising from the coupling to matter.

These relations may be important for computing measurable outputs of a cavity QED material. For example, if we probe the cavity-matter system through the transmissivity of the former, the transmitted signal is proportional to the dynamical response of the system. This is in turn related to two point correlators of bosonic fields that can be directly computed from relation (46) or analogues.

## 5 Testing the validity of the effective theory

Effective theories are a common tool in theoretical physics. To contextualize ours, we compare it, analytically and numerically, against other common ones in the fields of condensed matter and quantum optics. Importantly, we also perform a numerical analysis of its finite-size scaling. Because other effective theories are model dependent and to facilitate the numerical analysis, our testbed here will be the Dicke model,

$$\hat{\mathcal{H}} = \frac{\omega_z}{2} \sum_j \hat{\sigma}_j^z + \omega_c \hat{a}^\dagger \hat{a} + \frac{\lambda}{\sqrt{N}} \sum_j \hat{\sigma}_j^x \left( \hat{a} + \hat{a}^\dagger \right). \tag{48}$$

In the thermodynamic limit, it is exactly solvable, with the free energy per site given by [57]

$$\begin{cases} -\beta f = \ln \left[ 2 \cosh \left( \beta \frac{\omega_z}{2} \right) \right] & \text{if} \quad \lambda < \lambda_c \quad \text{or} \quad \beta > \beta_c, \\ -\beta f = \ln \left[ 2 \cosh \left( \beta \frac{4\lambda^2}{\omega_c} \sigma \right) \right] - \beta \frac{4\lambda^2}{\omega_c} \sigma^2 + \beta \frac{\omega_c \omega_z^2}{16\lambda^2} & \text{if} \quad \lambda > \lambda_c \quad \text{and} \quad \beta < \beta_c, \end{cases} \tag{49}$$

where $\lambda_c$ is given by $4\lambda_c^2 = \omega_c \omega_z$, $\beta_c$ is given by $\omega_c \omega_z = 4\lambda^2 \tanh(\beta_c \omega_z / 2)$ provided $\lambda > \lambda_c$ and $\sigma$ is found by solving $2\sigma = \tanh(4\beta\lambda^2 \sigma / \omega_c) \neq 0$. The Dicke model, besides being a well-known toy model, has the added benefit of being numerically amenable, by virtue of conserving the total spin $\hat{\mathbf{S}}^2 = \hat{S}_x^2 + \hat{S}_y^2 + \hat{S}_z^2$. Where $\hat{S}_\nu = \frac{1}{2} \sum_j \hat{\sigma}_j^\nu$. The numerical diagonalization can be done on each spin $S$ sector separately, with the total partition function given by [24]

$$Z = \sum_{S=s_0}^{N/2} \Omega(S, N) Z_S, \tag{50}$$

where $s_0 = 0$ (1/2) if $N$ is even (odd) and

$$\Omega(S, N) = \frac{N!(2S+1)}{(N/2 - S)!(N/2 + S + 1)!}. \tag{51}$$

Using total spin operators, the Hamiltonian can be rewritten as

$$\hat{\mathcal{H}} = \omega_z \hat{S}_z + \omega_c \hat{a}^\dagger a + 2g\hat{S}_x \left( \hat{a} + \hat{a}^\dagger \right), \tag{52}$$

with $g = \lambda / \sqrt{N}$. The corresponding effective Hamiltonian, according to our theory, reads

$$\hat{\mathcal{H}}_{\text{eff}} = \omega_z \hat{S}_z - \frac{4g^2}{\omega_c} \hat{S}_x^2. \tag{53}$$

A useful tool to facilitate the numerical diagonalization of the full Dicke model is the Polaron transformation. The Polaron transformation is given by $\hat{U}_{\mathrm{P}} = \exp\left(-\hat{\alpha}\hat{S}_x\right)$ with $\hat{\alpha} = 2\zeta(\hat{a}^\dagger - \hat{a})$ and $\zeta = g/\omega_c$. The resulting effective Hamiltonian $\hat{\mathcal{H}}_{\mathrm{P}} = \hat{U}_{\mathrm{P}}^\dagger \hat{\mathcal{H}} \hat{U}_{\mathrm{P}}$ reads

$$\hat{\mathcal{H}}_{\mathrm{P}} = \omega_z \left(\hat{S}_z \cosh\hat{\alpha} - i\hat{S}_y \sinh\hat{\alpha}\right) + \omega_c \hat{a}^\dagger a - \frac{4g^2}{\omega_c}\hat{S}_x^2. \tag{54}$$

The Polaron frame is particularly well-suited for exact diagonalization because the displacement of the bosons by $\hat{S}_x$ cancels the finite expectation value of $\hat{a}$, $\hat{a}^\dagger$ in the ordered phase. As a result, the number of bosons in the Polaron frame is only dependent on the temperature and the bosonic state-space can be truncated to a much smaller excitation cutoff than would otherwise be required [19,58]. This, paired with the conservation of the total spin, allows one to reach system sizes of $N \sim 100$. As we show below, this is effectively "close" to the thermodynamic limit. For this reason, we use the Hamiltonian in the Polaron frame (54) to do exact diagonalization of the full Dicke model.

## 5.1 Comparison with other effective theories

First, let us discuss the taxonomy of effective theories. We can distinguish two families: Hamiltonian theories, with unitary dynamics; and descriptions based on master equations and non-Hermitian Hamiltonians, with dissipative dynamics [59]. Because our theory belongs to the former, we will only consider other Hamiltonian theories in the comparison. We can also distinguish between perturbative and non-perturbative theories. In this regard, it must be noted that our effective Hamiltonian was derived in the $N \to \infty$ limit. As such, it can be regarded as a zeroth order perturbation theory with $1/N$ as the perturbative parameter. The extension to a full-fledged perturbation theory that can be taken to any order in $1/N$ is a work in progress. However, in this context we will use the term perturbative to refer only to parameters that affect the light-matter coupling regime, such as the light-matter coupling strength or the detuning between the cavity frequency and the characteristic energy scale of the matter subsytem. It is in this sense that our theory is non-perturbative. The simplest example of a perturbative theory would be standard perturbation theory on the light-matter coupling. To yield an effective Hamiltonian with perturbation theory, a separation of energy scales (large detuning) is required, with the more energetic sector of the spectrum mediating effective interactions on a low-energy subspace [60, 61]. A less cumbersome way to obtain the same effective description is by means of a Schrieffer-Wolff unitary transformation [62]. We will thus use the latter in our comparison. Regarding non-perturbative alternatives to our effective theory, it is interesting to consider the Polaron transformation itself [19]. For the Dicke model, $\left\langle \hat{a}^\dagger - \hat{a}\right\rangle$ vanishes in the thermodynamic limit, so $\hat{\alpha}$ in Eq. (54) vanishes as well. This means that, in the $N \to \infty$ limit, light and matter decouple in the Polaron Hamiltonian (54) and it coincides with our effective Hamiltonian (53) .

The aforementioned methods, namely the Schrieffer-Wolff and Polaron transformations, have the downside of being model-dependent. This is because the application of the unitary tranformations onto the original Hamiltoninan requires knowledge about the commutation relation between the matter part of the Hamiltonian and the matter part of the transformations. The matter subsystem must be specified a priori because knowledge of its algebra is required to obtain the effective theory. In contrast, our effective theory was obtained in a model-independent fashion. In all fairness, standard perturbation theory in the fast-cavity limit can also yield an effective matter model without knowledge of the matter subsystem. But in general, perturbation theory is model dependent [61].

For the Dicke model, the Schrieffer-Wolf transformation is given by $\hat{U}_{\mathrm{SW}} =$

$\exp\left(-\xi_- \hat{\mathcal{X}}_- - \xi_+ \hat{\mathcal{Y}}_-\right)$, with $\xi_\pm = g/(\omega_c \pm \omega_z)$ and

$$\hat{\mathcal{X}}_\pm = \hat{S}_- \hat{a}^\dagger \pm \hat{S}_+ \hat{a}\,, \tag{55}$$

$$\hat{\mathcal{Y}}_\pm = \hat{S}_+ \hat{a}^\dagger \pm \hat{S}_- \hat{a}\,. \tag{56}$$

The resulting effective Hamiltonian $\hat{\mathcal{H}}_{\mathrm{SW}} = \hat{U}_{\mathrm{SW}}^\dagger \hat{\mathcal{H}} \hat{U}_{\mathrm{SW}}$ can be expanded perturbatively and truncated at first order in $\xi_\pm$, yielding

$$\hat{\mathcal{H}}_{\mathrm{SW}} \approx \omega_z \hat{S}_z + \omega_c \hat{a}^\dagger a - \frac{2g^2 \omega_z}{\omega_c^2 - \omega_z^2} \hat{S}_z \left(\hat{a} + \hat{a}^\dagger\right)^2 - \frac{4g^2 \omega_c}{\omega_c^2 - \omega_z^2} \hat{S}_x^2\,. \tag{57}$$

Consequently, the parameters $\xi_\pm$ must be small, which requires a large detuning $|\omega_c - \omega_z| \gg g$.

In the fast-cavity limit $\omega_c \gg \omega_z$, the Hamiltonian simplifies to

$$\hat{\mathcal{H}}_{\mathrm{SW}} \approx \omega_z \hat{S}_z + \omega_c \hat{a}^\dagger \hat{a} - \frac{4g^2}{\omega_c} \hat{S}_x^2\,. \tag{58}$$

The spin and the cavity decouple and the matter part coincides with $\hat{\mathcal{H}}_{\mathrm{eff}}$. Taking the route of standard perturbation theory in this regime of detuning, one arrives at $\hat{\mathcal{H}}_{\mathrm{eff}}$ as well.

The slow-cavity regime is rather nuanced. For starters, there is no light-matter decoupling in this limit

$$\hat{\mathcal{H}}_{\mathrm{SW}} \approx \omega_z \hat{S}_z + \omega_c \hat{a}^\dagger a + \frac{2g^2}{\omega_z} \hat{S}_z \left(\hat{a} + \hat{a}^\dagger\right)^2\,. \tag{59}$$

In addition, the resulting model is ill-defined. The Hamiltonian commutes with $\hat{S}_z$, so one can fix the quantum number $m_z$, defined as $\hat{S}_z |S, m_z\rangle = m_z |S, m_z\rangle$. The Hamiltonian for an $m_z$ sector is quadratic in bosonic operators and can be diagonalized with a Bogoliubov transformation, $\hat{\mathcal{H}}_{\mathrm{SW}}(m_z) = \epsilon(m_z)\hat{b}^\dagger \hat{b}$. The resulting boson has a frequency

$$\epsilon(m_z) = \sqrt{\omega_c^2 + \frac{8g^2 \omega_c}{\omega_z} m_z}\,, \tag{60}$$

that vanishes for $m_z = -S$ at the critical coupling of the Dicke model ($4\lambda_c^2 = \omega_c \omega_z$) and is imaginary thereafter. The vanishing is a proper signature of the phase transition but becoming imaginary after is a symptom of the Hamiltonian being ill-defined. This is prevented by keeping the $\hat{S}_x^2$ term of $\hat{\mathcal{H}}_{\mathrm{SW}}$ that was neglected in passing from Eq. (57) to Eq. (59). Thus, in the slow-cavity limit the Schrieffer-Wolff Hamiltonian does not admit simplifications nor does it coincide with our effective Hamiltonian. This is a manifestation of the underlying perturbation theory, which requires a separation of energy scales that is unattainable when the cavity is taken as the less energetic subsystem, because its number of excitations (its energy) its unbounded. Additionally, the regime of resonant cavity and spins is not accessible to the Schrieffer-Wolff theory. These differences are numerically confirmed in Fig. 2 for the free energy per site of the Dicke model at high and low temperatures. We observe that the two theories agree in the fast-cavity regime, as expected, and deviate in the slow-cavity regime. Importantly, it is the Schrieffer-Wolff theory that deviates significantly from the full Dicke model past the critical point, whereas our effective theory shows the correct asymptotic behaviour. Let us emphasize again that the Schrieffer-Wolff theory is being used as a convenient proxy for standard perturbation theory.

The Schrieffer-Wolff Hamiltonian (57) requires a larger cutoff than the Polaron Hamiltonian, specially in the slow-cavity regime. For this reason, the system size has been kept at a conservative $N = 30$ for the comparison in Fig. 2 and the bosonic cutoff has been adjusted to reach convergence of the free energy. In practice, this meant a cutoff of $N_{\mathrm{ph}} = 100$ in the worst case.

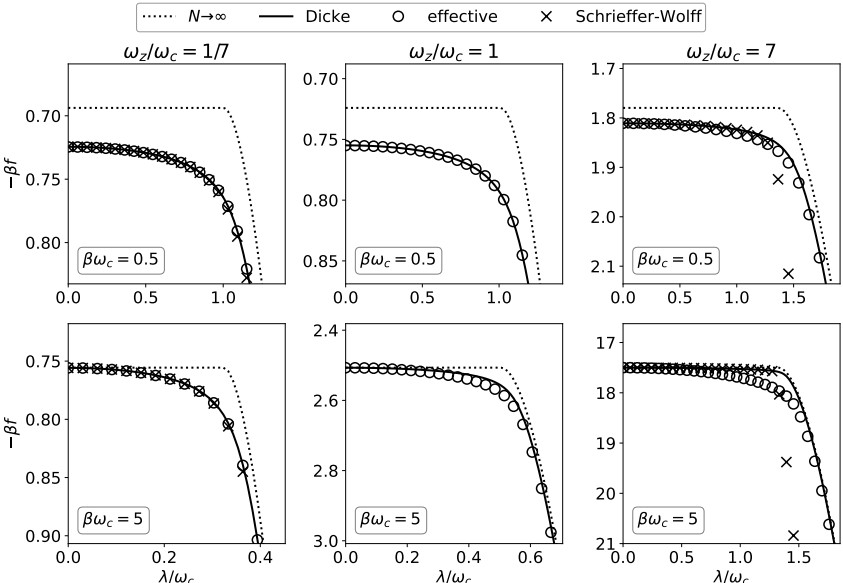

Figure 2: Comparison of the free energy per site computed by exact diagonalization of our effective Hamiltonian, exact diagonalization of the Schrieffer-Wolff Hamiltonian, exact diagonalization of the Dicke model in the Polaron frame and the exact analytical solution in the thermodynamic limit. The system size is $N = 30$ and the cutoff is $N_{\mathrm{ph}} = 100$ throughout.

## 5.2 Finite $N$ and approach to the thermodynamic limit

For an effective Hamiltonian that is only exact in the thermodynamic limit, it is interesting to test its finite-size scaling. We continue the numerical analysis on the Dicke model and in Fig. 3 we focus solely on our effective theory, which allows us to lower the cutoff to $N_{\mathrm{ph}} = 10$ and explore much larger system sizes. We distinguish three regimes of fast cavity, resonance and slow cavity and find that each presents a characteristic finite-size scaling. In the fast cavity limit, the effective theory quickly converges to the full Dicke model, already for $N = 30$, and then they both slowly converge to the $N \to \infty$ analytical solution later, with relative differences of $\sim 2\%$ at the critical point for $N = 150$. As one moves towards the slow-cavity regime, the convergence of the effective theory to the full Dicke model slows down, whereas the convergence of the full Dicke model to the $N \to \infty$ analytical solution accelerates. Here we have focused in the free energy at low temperature, which for $\beta \omega_c = 5$ is practically equivalent to the ground-state energy, because it is the regime in which the effective theory is most challenged to meet the full Dicke model. In contrast, at high temperature convergence is obtained much faster, akin to the fast-cavity regime at low temperature, as can be seen in the top row of Fig. 2 already at $N = 30$. In summary, we find that the effective theory scales well with $N$, with relative differences with the full Dicke model $< 1\%$ at the critical point for $N \sim 100$ in every regime. In the high temperature and/or fast-cavity regime, the convergence is even faster, ocurring at $N \sim 30$.

These numerical results imply an important advantage of our effective theory in regards to its applicability to any light-matter coupling regime, hence the surname non-perturbative. They also prove that it is not equivalent to standard perturbation theory. As a caveat, we acknowledge that the theory has only been numerically compared and tested in the Dicke model, although we expect the theory to be equally applicable to other systems. As a matter of fact, in the next section we employ the theory to rederive and generalize results in other models.

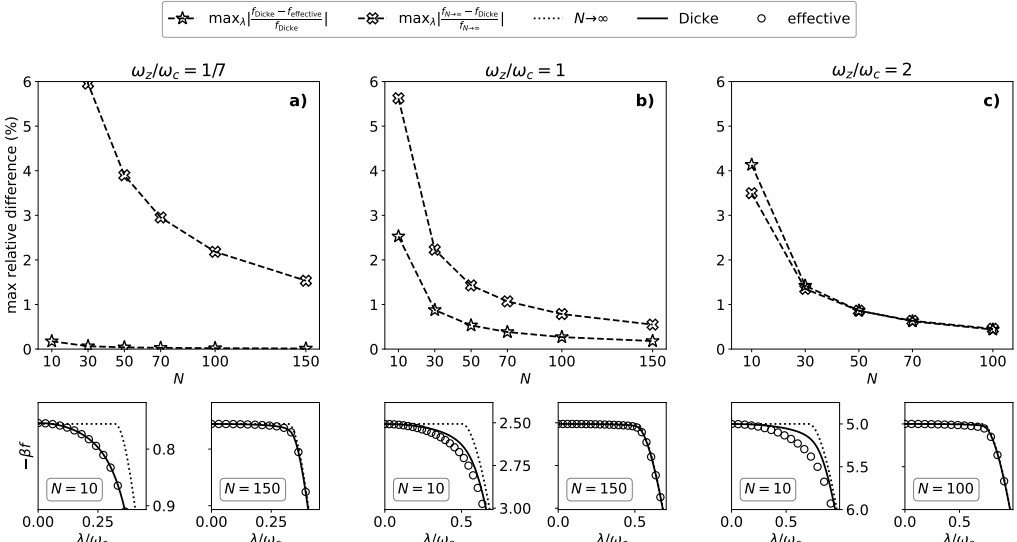

Figure 3: Finite-size scaling of the free energy per site computed with our effective theory. Top: Maximum relative difference between the free energies computed by exact diagonalization of our effective Hamiltonian, exact diagonalization of the Dicke model in the Polaron frame and the exact analytical solution in the thermodynamic limit. Bottom: Plots of the free energy at small and large system sizes. a) In the fast-cavity regime $\omega_z/\omega_c = 1/7$. b) At resonance $\omega_z/\omega_c = 1$. c) In the slow-cavity regime $\omega_z/\omega_c = 2$. For reasons of numerical intractability, values for $N = 150$ were unattainable in the slow-cavity regime. The temperature is $\beta\omega_c = 5$ and the cutoff is $N_{\mathrm{ph}} = 10$ throughout.

## 6 Applications

Here, we showcase the effectiveness of the formalism developed in Section 4 in the study of several models that have received recent attention in the literature, namely: equilibrium superradiance (photon condensation) with electric coupling, the 2D free electron gas in a cavity, and the modification of interactions in magnetic materials coupled to a cavity (magnetic cavity QED).

### 6.1 Equilibrium superradiance (photon condensation) with electric coupling

Despite its long history, the problem of the existence (or lack thereof) of a superradiant phase transition when an ensemble of atoms are immersed in a cavity has attracted renewed attention. After 50 years from its original theoretical description, the transition has not been experimentally measured [63] and there is vivid debate to elucidate the reason. So far, the understanding is as follows. If one adopts the correct gauge-invariant description of a single uniform cavity mode, the phase transition, defined as the appearance of a finite population of transverse photons, is forbidden. This has been proven true even without the dipole approximation [28, 32, 37, 49]. This transition is often referred to as *equilibrium superradiance* or *photon condensation*, although some ambiguity still surrounds these terms [36]. Moreover, critical conditions have been obtained in the case of non-uniform cavity fields [29–31]. Subtleties arise from the fact that the separation between light and matter subsystems is manifestly gauge-dependent. For that reason, we prefer to characterize the phase transition in terms of gauge invariant observables such as the transverse photon number. However, although theoretically valid, this choice suffers from the fact that not all observables, in particular the

transverse photon number, are experimentally measurable, a handicap that challenges their validity as signatures of a phase transition. To avoid this pitfall, we propose to look at this issue from a different perspective: instead of arguing whether the phase transition has a "light" or a "matter" signature, we will simply compare the behaviour of the material when it is placed outside a cavity, i.e. in a region of space with an unconfined electromagnetic field, against its behaviour when it is placed inside a cavity, i.e. in a region of space where the electromagnetic field is confined. Note that this is *the experimentally relevant question*. In fact, alterations of matter and condensation of photons are two sides of the same coin.

Our formalism is particularly well suited to answer this question because it eliminates the cavity, reformulating the model in terms of the original (in free space) matter Hamiltonian plus an effective interaction mediated by the cavity. Whether or not a material behaves differently after placing it inside a cavity will be answered by studying the impact, or lack thereof, that the effective coupling has in its behaviour.

We center on the study on a matter model such as the one in Eq. 17, where the Zeeman coupling to the electron spins is neglected. Accordingly, we have $c_k = \lambda_k^{-1/2} c_k^e$ and

$$\hat{\mathcal{C}}_\kappa = \sum_j \sqrt{\frac{1}{m\omega_k}} \hat{p}_j \hat{u}_\kappa(\hat{r}_j). \tag{61}$$

Then, in the thermodynamic limit, the effective matter Hamiltonian, according to Eq. 41, is

$$\hat{\mathcal{H}}_{\text{eff}} = \hat{\mathcal{H}}'_M - \sum_\kappa \frac{(c_k^e)^2}{\omega_k + 4\Delta_k} \sum_{ij} \frac{1}{m\omega_k} (\hat{p}_j \hat{u}_\kappa(\hat{r}_j))(\hat{u}_\kappa^\dagger(\hat{r}_i)\hat{p}_i). \tag{62}$$

We have an effective all-to-all coupling between the electrons' momenta. The strength of the coupling between a pair of electrons depends on their positions. We treat here the case of the electron gas, but the results that follow extend to the case of localized atoms as they are independent of whether the EM fields depend on the position of the electrons dynamically or parametrically.

### 6.1.1  Uniform cavity fields

If we make the approximation to a setup with a single uniform mode $k \to 0$, $\sigma = 1$, we end up with

$$\hat{\mathcal{H}}_{\text{eff}} = \hat{\mathcal{H}}'_M - \frac{(c_0^e)^2}{\omega_0 + 4\Delta_0} \sum_{ij} \frac{1}{m\omega_0} (e_1 \hat{p}_j)(e_1 \hat{p}_i), \tag{63}$$

where $e_1$ is the polarization vector of the electromagnetic mode. Now that the spatial dependence of the coupling is no longer present, the coupling is uniform and all-to-all, a situation that can be treated exactly with a mean field theory in which $\hat{p}_j = \langle \hat{p} \rangle + \delta \hat{p}_j$.

$$
\begin{aligned}
\hat{\mathcal{H}}_{\text{eff}} &= \hat{\mathcal{H}}'_M - 2\frac{4\Delta_0}{\omega_0 + 4\Delta_0} \sum_j \frac{(e_1 \hat{p}_j)\langle e_1 \hat{p}\rangle}{2m} + N\frac{4\Delta_0}{\omega_0 + 4\Delta_0}\frac{\langle e_1 \hat{p}\rangle^2}{2m} \\
&= \sum_j \frac{1}{2m}\left(\hat{p}_j - \frac{4\Delta_0}{\omega_0 + 4\Delta_0} e_1 \langle e_1 \hat{p}\rangle\right)^2 + \hat{\mathcal{H}}'_{qq} + \hat{\mathcal{V}} + N\frac{4\Delta_0\omega_0}{(\omega_0 + 4\Delta_0)^2}\frac{\langle e_1 \hat{p}\rangle^2}{2m}.
\end{aligned} \tag{64}
$$

For convenience, we have expressed the electric coupling $c_0^e$ in terms of $\Delta_0$. Now we observe that

$$\hat{p}_j - \frac{4\Delta_0}{\omega_0 + 4\Delta_0} e_1 \langle e_1 \hat{p}\rangle = [\hat{r}_j, \hat{\mathcal{H}}_{\text{eff}}], \tag{65}$$

and since $\langle[\hat{r}_j, \hat{\mathcal{H}}_{\text{eff}}]\rangle = 0$, we have

$$\langle e_1 \hat{p} \rangle - \frac{4\Delta_0}{\omega_0 + 4\Delta_0} \langle e_1 \hat{p} \rangle = 0. \tag{66}$$

Note that this condition is only satisfied if $\langle e_1 \hat{p} \rangle = 0$. This is because

$$\frac{4\Delta_0}{\omega_0 + 4\Delta_0} \leq 1. \tag{67}$$

In fact, one could express the critical condition as $\omega_0 + 4\Delta_0 = 4\Delta_0$, which is clearly never satisfied. Consequently, we find that $\hat{\mathcal{H}}_{\text{eff}} = \hat{\mathcal{H}}'_M$, which serves as proof that the effect of the transverse cavity fields on the matter system is non-existent. If there is a phase transition, its origin lies within $\hat{\mathcal{H}}'_M$. Meaning that it is either caused by intrinsic interactions in the material $\hat{\mathcal{H}}_M$ or induced by electrostatic effects arising from the interaction between real charges and image charges.

To reinforce this result, let us compare it with the case in which the diamagnetic term is omitted in the light-matter interaction. This equates to setting $\Delta_0 = 0$. Of course, one must then be careful not no express the electric coupling $c_0^e$ in terms of $\Delta_0$ as we have done before, otherwise we cannot zero $\Delta_0$ independently. Then, the critical condition becomes

$$\frac{N(c_0^e)^2}{\omega_0^2} = 1, \tag{68}$$

which can be satisfied if the coupling $c_0^e$ is large enough. We have thus not only arrived to the well-known no-go theorem for single mode cavities, but we are also able to explain the presence of critical behaviour when the diamagnetic term is ignored. We must also emphasize that the results obtained here are valid at any temperature and that no assumptions have been made in the derivation regarding the order of the phase transition. So the result is a no-go theorem for both continuous and discontinuous phase transitions. This generalizes the works of Refs. [28, 49] that discuss the zero temperature case of second and first order quantum phase transitions.

### 6.1.2 Non-uniform cavity fields

Let us return now to the multimode case described by Eq. (62). For finite, but low-enough temperatures, minimizing the free energy is equivalent to minimizing the energy (See SM from [22]). In that case, a phase transition will occur if the condition $\text{Tr}_M(\hat{\mathcal{H}}_{\text{eff}}\rho) \leq \text{Tr}_M(\hat{\mathcal{H}}_{\text{eff}}\rho_0)$ is met, where $\rho_0$ is the density matrix describing the thermal ensemble given by $\hat{\mathcal{H}}_M$ and $\rho$ is a tentative density matrix describing the thermal ensemble given by $\hat{\mathcal{H}}_{\text{eff}}$. The critical condition can be written as

$$\text{Tr}_M(\hat{\mathcal{H}}_M\rho) - \text{Tr}_M(\hat{\mathcal{H}}_M\rho_0) \leq \sum_\kappa \frac{c_k^2}{\tilde{\omega}_k} \text{Tr}_M(\hat{\mathcal{C}}_k\hat{\mathcal{C}}_k^\dagger \rho) - \sum_\kappa \frac{c_k^2}{\tilde{\omega}_k} \text{Tr}_M(\hat{\mathcal{C}}_k\hat{\mathcal{C}}_k^\dagger \rho_0). \tag{69}$$

Making use of the relation between light and matter observables (46) this can be reexpressed as

$$\text{Tr}_M(\hat{\mathcal{H}}_M\rho) - \text{Tr}_M(\hat{\mathcal{H}}_M\rho_0) \leq \sum_\kappa \tilde{\omega}_k \langle b_\kappa^\dagger b_\kappa \rangle. \tag{70}$$

Close to the phase transition, the left term can be expressed in terms of $\langle b_\kappa^\dagger b_\kappa \rangle$ using the stiffness theorem [64]. In particular, we can consider the zero temperature case, where the

critical condition is written in terms of $|\psi\rangle$ the tentative matter ground state, and $|\psi_0\rangle$ the matter ground state in the absence of light-matter coupling

$$\langle\psi|\hat{\mathcal{H}}_M|\psi\rangle - \langle\psi_0|\hat{\mathcal{H}}_M|\psi_0\rangle \leq \sum_\kappa \tilde{\omega}_{\boldsymbol{k}} \left\langle b_\kappa^\dagger b_\kappa \right\rangle. \tag{71}$$

Moreover, we are not restricted to considering only electric coupling as in Eq. (61), the Zeeman coupling can be included in $\hat{\mathcal{C}}_\kappa$ as well, either on its own or combined with the electric coupling. With these extension, we are able to recover the full results of Ref. [31], demonstrating that our formalism is able to provide a complete description of the phenomenon of photon condensation.

## 6.2 The 2D free electron gas in a cavity

The free electron gas is a paradigmatic model in condensed matter physics. It was originally formulated by Sommerfeld [65] for the study of the thermal and electronic properties of materials, and it later became the building block of the Fermi liquid theory, developed by Landau [64,66]. Its interacting generalization, known as the homogeneous electron gas or jellium model, is the basis for advanced computational techniques such as density functional theory (DFT) and the local density approximation (LDA) [67,68]. Lowering the dimensionality, 2D electron gases can be found in semiconductor devices in solid state physics [69]. They also describe the integer quantum Hall effect when subjected to strong magnetic fields and low temperature [70,71]. Having established that the electron gas is an interesting model exhibiting rich phenomenology, it seems suggestive to explore the extent to which its behaviour can be altered by coupling it to a cavity. The first steps in this direction have already experimentally confirmed the breakdown of topological protection by cavity vacuum fields [72], and predicted the existence of a polaritonic Hofstadter butterfly and modified integer Hall conductance [73] and cavity-mediated hopping [74], in the integer quantum Hall effect.

Ref. [50] is dedicated to the exact analytical solution of the 2D free electron gas in a cavity, making it the perfect testbed for our effective theory. In this subsection we recover their results straightforwardly with our formalism. To adhere to the same assumptions as the authors, we will consider a multimode description of the cavity fields, but with the peculiarity that the wavelength of every mode be large enough that the electrons see the mode as uniform. In that case we have $\hat{\boldsymbol{u}}_\kappa(\hat{\boldsymbol{r}}_j) \equiv \boldsymbol{e}_\sigma$ where $\boldsymbol{e}_\sigma$ is the polarization vector of the mode, but we retain the summation in $\kappa$ when summing over modes. In addition, the mode-mode interactions arising from the diamagnetic term are completely disregarded. On the matter side, free electrons are simply described by the Hamiltonian

$$\hat{\mathcal{H}}_M = \sum_j \frac{\hat{\boldsymbol{p}}_j^2}{2m}, \tag{72}$$

where the 2D nature of the system is reflected in the fact that the momenta are confined to a plane: $\hat{\boldsymbol{p}}_j = (\hat{p}_{j,x}, \hat{p}_{j,y})$. So the complete Hamiltonian for the system is given by

$$\hat{\mathcal{H}} = \hat{\mathcal{H}}_M + \sum_\kappa \omega_{\boldsymbol{k}} \hat{a}_\kappa^\dagger \hat{a}_\kappa + \sum_j \sum_\kappa \frac{q}{m}(\hat{\boldsymbol{p}}_j \boldsymbol{e}_\sigma) A_{\boldsymbol{k}} \left(\hat{a}_\kappa + a_\kappa^\dagger\right) + \sum_j \frac{q^2}{2m} \sum_\kappa A_{\boldsymbol{k}}^2 \left(\hat{a}_\kappa + \hat{a}_\kappa^\dagger\right)^2. \tag{73}$$

Note here that image charges from the cavity boundaries are not included in the description. The terms quadratic in photon operators can be brought to diagonal form by a Bogoliubov transformation

$$\hat{a}_{\boldsymbol{k},\sigma}^\dagger = \cosh(\theta_{\boldsymbol{k}})\hat{b}_{\boldsymbol{k},\sigma}^\dagger - \sinh(\theta_{\boldsymbol{k}})\hat{b}_{\boldsymbol{k},\sigma}, \tag{74}$$

where $\cosh(\theta_k) = (\lambda_k + 1)/(2\sqrt{\lambda_k})$, $\sinh(\theta_k) = (\lambda_k - 1)/(2\sqrt{\lambda_k})$ and $\lambda_k = \sqrt{1 + 4\Delta_k/\omega_k}$. The resulting Hamiltonian is

$$\hat{\mathcal{H}} = \hat{\mathcal{H}}_M + \sum_\kappa \omega_k \lambda_k \hat{a}_\kappa^\dagger \hat{a}_\kappa + \sum_j \sum_\kappa c_k^e \lambda_k^{-1/2} \sqrt{\frac{1}{m\omega_k}} (\hat{p}_j e_\sigma)(\hat{b}_\kappa + \hat{b}_\kappa^\dagger). \tag{75}$$

At this point, the paralelism with Eq. 17 is complete by simply setting $c_k = c_k^e \lambda_k^{-1/2}$ and $\hat{\mathcal{C}}_\kappa = \hat{\mathcal{C}}_\kappa^\dagger = \sqrt{\frac{1}{m\omega_k}} e_\sigma \hat{P}$, with $\hat{P} = \sum_j \hat{p}_j$. Consequently, the effective Hamiltonian is given by

$$\hat{\mathcal{H}}_{\text{eff}} = \sum_j \frac{\hat{p}_j^2}{2m} - \sum_\kappa \frac{(c_k^e)^2}{\tilde{\omega}_k \lambda_k} \frac{1}{m\omega_k} (e_\sigma \hat{P})^2 = \frac{1}{2m} \left[ \sum_j \frac{\hat{p}_j^2}{2m} - \left( \sum_k \frac{4\Delta_k}{\omega_k + 4\Delta_k} \right) \frac{1}{N} \sum_\sigma (e_\sigma \hat{P})^2 \right]. \tag{76}$$

The reader should note that this Hamiltonian corresponds precisely with the ground-state energy obtained in [50]. We find the added benefit that the Hamiltonian offers more flexibility in the amount of calculations that can be done from here. For instance finite temperature calculations, in contrast to the ground-state energy, which is limited in that regard.

## 6.3 Magnetic cavity QED

As a final example we explore how to modify the magnetic properties of materials. Ferromagnetic enhancement by collective strong coupling to a cavity has already been demonstrated experimentally with oxide nanoparticles [23], in connection with the modification of superconducting properties [18]. Here, instead, we consider molecular nanomagnets. The vast majority of them are neutral and present a near zero electric dipole. Their behaviour is then accurately described by an effective "giant"-spin Hamiltonian $\hat{\mathcal{H}}_S$, that includes the effect of magnetic anisotropy and the Zeeman coupling $\hat{\mathcal{H}}_Z = -g\mu_B \hat{S} \cdot \hat{B}$ to a magnetic field. With spin operators $[S_i, S_j] = i\epsilon_{ijk} S_k$. These molecules can be coupled to superconducting cavities [75] and have been recently studied as a good candidate to achieve equilibrium condensation, and beyond, to achieve the modification of the ferromagnetism of materials from the coupling to electromagnetic cavities [22]. To study the situation where an ensemble of such molecular spins is placed within a cavity, we will consider that $\hat{\mathcal{H}}_S$ only contains the coupling to external classical fields, and treat separately the coupling of the spins to the quantized magnetic field of an electromagnetic cavity. The resulting full Hamiltonian reads [76]

$$\hat{\mathcal{H}} = \hat{\mathcal{H}}_S' + \hat{\mathcal{H}}_{\text{EM}} - g\mu_B \sum_j \hat{S}_j \hat{B}(r_j). \tag{77}$$

We write $\hat{\mathcal{H}}_S'$ to indicate that the effect of image spins from the cavity boundaries can be taken into account, modifying the behaviour of the spin ensemble beyond the Zeeman coupling. Owing to the fact that the molecules are completely described by their spin degrees of freedom $\hat{S}_j$, their positions are assumed fixed at positions $r_j$. The intensity of the cavity field at each position will determine the strength of the individual couplings. In contrast to previous examples, the positions $r_j$ act simply as parameters of the model and not as operators in a Hilbert space, they do not constitute a dynamical degree of freedom. Introducing the explicit forms of the electromagnetic Hamiltonian and magnetic field, the Hamiltonian reads

$$\hat{\mathcal{H}} = \hat{\mathcal{H}}_S + \sum_\kappa \omega_k \hat{a}_\kappa^\dagger \hat{a}_\kappa - g\mu_B \sum_j \sum_\kappa \hat{S}_j B_k (u_{\perp,\kappa}(r_j) \hat{a}_\kappa + h.c.). \tag{78}$$

Identifying the couplings $c_k = g\mu_B B_k$ and the coupling operators

$$\hat{\mathcal{C}}_\kappa = \sum_j \hat{S}_j u_{\perp,\kappa}(r_j), \tag{79}$$

we arrive to the effective Hamiltonian

$$\hat{\mathcal{H}}_{\text{eff}} = \hat{\mathcal{H}}_S - \sum_{\kappa} \frac{c_k^2}{\omega_k} \sum_{ij} \left(\hat{\boldsymbol{s}}_i \boldsymbol{u}_{\perp,\kappa}(\boldsymbol{r}_i)\right)\left(\hat{\boldsymbol{s}}_j \boldsymbol{u}_{\perp,\kappa}^*(\boldsymbol{r}_j)\right). \tag{80}$$

This is a generalization to an arbitrary cavity geometry of the effective Hamiltonian presented in [22]. We can see that the degree to which two distant spins couple depends on the amplitude of the transverse mode functions at each position, but also on the degree to which each spin aligns with the local magnetic field. This is a trait inherited from the original Hamiltonian (77), where we can see that the energy is lowered when all the spins are parallel to the local magnetic field. The extent to which this local interaction with the magnetic field will translate to an effective ferromagnetic interaction among the spins is dictated by the spatial uniformity of the magnetic field. Only if we consider the simplest case of a uniform single-mode field $\boldsymbol{k} \to 0$, $\sigma = 1$ will we have an effective ferromagnetic all-to-all interaction between spins. This situation is only an ideal limit-case of the more general picture in which some degree of non-uniformity is unavoidable. This caveat is particularly important because, as explained in [22], in order to achieve significant modifications of the behaviour of the spin system, the filling factor -a proxy for the degree to which the magnetic molecules occupy the region of space where the field has a significant intensity- must be maximized. If we consider a molecular crystal, there is a clear trade-off between increasing the crystal size to cover a larger portion of the cavity volume and keeping the crystal within a small-enough region of space that the experienced field is uniform.

# 7 Conclusion and outlook

We have established an operational way to understand how matter is modified when it is immersed in a cavity. We have explicitly isolated the effect of quantum fluctuations of light. Taking advantage of the fact that the EM field is a free bosonic field, we have traced it our exactly with the coherent path integral formalism to obtain an effective theory of matter. Our theory comes in the form of an effective action containing only matter degrees of freedom, where the effect of light is given by a non-local kernel. In the large-$N$ limit, this kernel becomes local and the theory can be written as a matter Hamiltonian containing effective matter-matter interactions. Our starting point is the minimal coupling of light and matter in an arbitrary geometry, so our results are quite general. Besides, our theory can be adapted to describe the coupling to other bosonic excitations, such as phonons, plasmons or magnons.

This effective Hamiltonian has been tested on the Dicke model. We have shown that it is more versatile than perturbative effective theories. Its ability to cover the full light-matter coupling regime merits its denomination as non-perturbative. This general theory has been in three examples of current relevance. First, photon condensation was understood from our effective theory. In particular, its non-existence when uniform cavity fields are considered and gauge invariance is respected, and the condition for its existence in the case of non-uniform cavity fields. Then, we recovered the exact solution for the 2D free electron gas in a cavity within our effective formalism. Finally, we studied a system of molecular spins in a cavity, laying the foundations of magnetic cavity QED. In the limit case of uniform fields, the effective interactions are ferromagnetic and all-to-all. Conversely, for general non-uniform fields, the effective spin-spin interactions are position dependent. This could be leveraged to engineer complex interactions between distant spins, benefiting from the different directions and intensities of the mode functions at different locations within the cavity. The appearance of all-to-all position-dependent interactions is not exclusive to magnetic systems, but instead intrinsic to the formalism, so the engineering of interactions, with applications in sight, can also

be explored in other systems. Unlike other coupling mechanisms, the intensity of the interaction between two constituents of a material is not inversely determined by their distance but instead depends on the intensity of the cavity field at the particles positions. If they both lie at antinodes of the cavity field, the coupling will be maximal irrespective of their distance, hinting at the possibility of inducing long-range interactions within materials. These have been studied to generate non-geometrical frustration, to induce quantum spin liquid phases [77]. Importantly enough, our treatment is exact and does not rely on perturbative expansions on the coupling, detuning, etc. Therefore, the effective Hamiltonian is as valid as the underlying modelization of the light-matter coupling. Thus, the tailored interactions can be used for, *e.g.* quantum simulators in the whole range of matter-matter interaction strengths [78–80]. On the other hand, the explicitness of the effective Hamiltonian provides a direct way to classify the obtained models, in terms of symmetries, topology, integrability, etc.

From a theoretical perspective, the theory may find several extensions. For instance by considering the coupling of matter to non-bosonic, e.g. fermionic or spin, degrees of freedom. Alternatively, staying within the realm of cavity QED, it might be illuminating to develop the dynamical counterpart of this equilibrium effective theory, using real time path integrals and linear response theory. Finally, a systematic treatment of the non-local kernel, in relation to bridging the gap from finite to infinite $N$, remains to be found.

## Acknowledgements

The authors acknowledge clarifying discussions with Peter Rabl, Adam Stokes, Ahsan Nazir, Pierre Nataf, Thierry Champel and Denis Basko. Also funding from the EU (QUANTERA SUMO and FET-OPEN Grant 862893 FATMOLS), the Spanish Government Grant PID2020-115221GB-C41/AEI/10.13039/501100011033, the Gobierno de Aragón (Grant E09-17R Q-MAD) and the CSIC Quantum Technologies Platform PTI-001.

## A   Light-matter Hamiltonian for a system of localized emitters

Let us now adapt the formulation of the gas model presented in Sec. 3 to the case of localized emitters. In the case of fixed emitters, such as atoms or molecules, the charged particles are electrons with $q = -e$, where $e$ is the electron's charge, the Hamiltonian is given by

$$\hat{\mathcal{H}}_M = \sum_j \frac{\hat{\boldsymbol{p}}_j^2}{2m} + \hat{\mathcal{H}}_{qq} + \hat{\mathcal{H}}_{qQ} + \sum_j \hat{v}(\hat{\boldsymbol{r}}_j + \boldsymbol{R}_j) + \sum_{i \neq j} \hat{V}(\hat{\boldsymbol{r}}_i + \boldsymbol{R}_i, \hat{\boldsymbol{r}}_j + \boldsymbol{R}_j). \tag{81}$$

Here, $\hat{\boldsymbol{r}}_j$ and $\hat{\boldsymbol{p}}_j$ are the relative position and momentum operators of the $j$-th electron with respect to its bounding core, $\boldsymbol{R}_j$ is the position of the core of the $j$-th electron. Note that several electrons can share the same core. The set of $\{\boldsymbol{R}_j\}$ constitute fixed parameters of each model. In this case $\hat{\mathcal{H}}_{qQ}$ constitutes the Coulombic interaction that bounds each electron to its respective core. Once again this matter Hamiltonian can be summarized as $\hat{\mathcal{H}}_M = \hat{\mathcal{T}} + \hat{\mathcal{H}}_{qq} + \hat{\mathcal{H}}_{qQ} + \hat{\mathcal{V}}$. After coupling this material subsystem to the quantized electromagnetic field in a cavity, the Hamiltonian for the complete system in the Coulomb gauge reads

$$\hat{\mathcal{H}} = \sum_j \frac{(\hat{\boldsymbol{p}}_j - q\hat{\boldsymbol{A}}(\boldsymbol{R}_j))^2}{2m} + \hat{\mathcal{H}}'_{qq} + \hat{\mathcal{H}}'_{qQ} + \hat{\mathcal{V}} + \hat{\mathcal{H}}_{\mathrm{EM}} - \frac{g\mu_B}{2} \sum_j \hat{\boldsymbol{\sigma}}_j \hat{\boldsymbol{B}}(\boldsymbol{R}_j). \tag{82}$$

Note that in contrast to the gas, the fact that the dynamic charges are bounded allows us to take the long-wavelength approximation, assuming that the electromagnetic field is constant at

each particle's location, such that $\hat{A}(\hat{r}_j + R_j) \approx \hat{A}(R_j)$, $\hat{B}(\hat{r}_j + R_j) \approx \hat{B}(R_j)$. The main difference with the description of the gas is that now the electromagnetic fields are operators acting solely on the Hilbert space of the light. The mode functions that describe their spatial dependence are not promoted to operators since they depend on the fixed positions $\{R_j\}$. After expanding the kinetic term and substituting the explicit forms of the vector potential and the magnetic field, we obtain

$$
\begin{aligned}
\hat{\mathcal{H}} = \hat{\mathcal{H}}'_M &+ \sum_\kappa \omega_k \hat{a}^\dagger_\kappa \hat{a}_\kappa - \sum_j \sum_\kappa \frac{q}{m} \hat{p}_j A_k \left( u_\kappa(R_j) \hat{a}_\kappa + h.c. \right) \\
&+ \sum_j \frac{q^2}{2m} \sum_{\kappa,\kappa'} A_k A_{k'} \left( u_\kappa(R_j) \hat{a}_\kappa + h.c. \right) \left( u_{\kappa'}(R_j) \hat{a}_{\kappa'} + h.c. \right) \\
&- \frac{g \mu_B}{2} \sum_j \sum_k \hat{\sigma}_j B_k \left( u_{\perp,\kappa}(R_j) \hat{a}_\kappa + h.c. \right).
\end{aligned}
\tag{83}
$$

To condense the notation, we define the quantities

$$
U_{\kappa,\kappa'} = N^{-1} \sum_j u_\kappa(R_j) u_{\kappa'}(R_j),
\tag{84}
$$

$$
\bar{U}_{\kappa,\kappa'} = N^{-1} \sum_j u_\kappa(R_j) u^*_{\kappa'}(R_j).
\tag{85}
$$

With these, we write the terms that depend quadratically on the bosonic operators as

$$
\hat{\mathcal{H}}_{\mathrm{ph}} = \sum_\kappa \omega_k \hat{a}^\dagger_\kappa \hat{a}_\kappa + \sum_{\kappa,\kappa'} \Delta_{k,k'} \left( U_{\kappa,\kappa'} \hat{a}_\kappa \hat{a}_{\kappa'} + \bar{U}_{\kappa,\kappa'} \hat{a}_\kappa \hat{a}^\dagger_{\kappa'} + h.c. \right).
\tag{86}
$$

Whereas in the case of a gas the photonic Hamiltonian could not be diagonalized in the general case, the fact that the coefficients of the quadratic terms are now c-numbers allows us to exactly diagonalize the photonic terms by means of a Bogoliubov transform. First, we define $\Psi^t = (\hat{a}^\dagger_{\kappa_1}, \ldots, \hat{a}^\dagger_{\kappa_M}, \hat{a}_{\kappa_1}, \ldots, \hat{a}_{\kappa_M})$, which allows us to write the Hamiltonian as

$$
\hat{\mathcal{H}}_{\mathrm{ph}} = \frac{1}{2} \Psi^\dagger \mathbb{H} \Psi, \qquad \text{with} \qquad \mathbb{M} = \begin{pmatrix} \mathbb{H}_1 & \mathbb{H}_2 \\ \mathbb{H}_2^\dagger & \mathbb{H}_1^* \end{pmatrix},
\tag{87}
$$

where the $M \times M$ blocks $\mathbb{H}_1$ and $\mathbb{H}_2$ are defined in terms of their matrix elements

$$
(\mathbb{H}_1)_{\kappa,\kappa'} = 2 \Delta_{\kappa,\kappa'} \bar{U}_{\kappa,\kappa'} + \delta_{\kappa\kappa'} \omega_k,
\tag{88}
$$

$$
(\mathbb{H}_2)_{\kappa,\kappa'} = 2 \Delta_{\kappa,\kappa'} U_{\kappa,\kappa'}.
\tag{89}
$$

The Bogoliubov transform is determined by a transformation matrix $\mathbb{T}$, obeying the pseudounitarity condition $\mathbb{T}^\dagger \mathbb{I}_- \mathbb{T} = \mathbb{I}_-$, which defines a new set of bosonic operators $\Phi = \mathbb{T}^{-1} \Psi = (\hat{b}^\dagger_{\kappa_1}, \ldots, \hat{b}^\dagger_{\kappa_M}, \hat{b}_{\kappa_1}, \ldots, \hat{b}_{\kappa_M})$ and a new Hamiltonian matrix $\mathbb{H} = \mathbb{T}^\dagger \mathbb{H} \mathbb{T}$ such that $\mathbb{H}$ is diagonal [52]. The resulting Hamiltonian can be written as

$$
\hat{\mathcal{H}}_{\mathrm{ph}} = \sum_\kappa \tilde{\omega}_k b^\dagger_\kappa b_\kappa.
\tag{90}
$$

The pseudounitarity of $\mathbb{T}$ ensures that the new bosonic operators obey the canonical commutation relations. The $\mathbb{I}_-$ block matrix is defined as

$$
\mathbb{I}_- = \begin{pmatrix} \mathbb{I}_M & 0 \\ 0 & -\mathbb{I}_M \end{pmatrix},
\tag{91}
$$

with $\mathbb{I}_M$ the identity matrix of dimension $M$. The old bosonic operators can be written in terms of the new ones as

$$\hat{a}_\kappa = \sum_{\kappa'} (\alpha_{\kappa,\kappa'} \hat{b}_{\kappa'} + \beta_{\kappa,\kappa'} \hat{b}_{\kappa'}^\dagger). \tag{92}$$

With this, and defining

$$\tilde{A}_{\boldsymbol{k}} \tilde{\boldsymbol{u}}_\kappa(\boldsymbol{R}_j) = \sum_{\kappa'} A_{\boldsymbol{k}'} \left( \boldsymbol{u}_{\kappa'}(\boldsymbol{R}_j) \alpha_{\kappa,\kappa'} + \boldsymbol{u}_{\kappa'}^*(\boldsymbol{R}_j) \beta_{\kappa,\kappa'}^* \right), \tag{93}$$

$$\tilde{B}_{\boldsymbol{k}} \tilde{\boldsymbol{u}}_{\perp,\kappa}(\boldsymbol{R}_j) = \sum_{\kappa'} B_{\boldsymbol{k}'} \left( \boldsymbol{u}_{\perp,\kappa'}(\boldsymbol{R}_j) \alpha_{\kappa,\kappa'} + \boldsymbol{u}_{\perp,\kappa'}^*(\boldsymbol{R}_j) \beta_{\kappa,\kappa'}^* \right), \tag{94}$$

and the constants

$$\tilde{c}_{\boldsymbol{k}}^e = q \tilde{A}_{\boldsymbol{k}} \sqrt{\frac{\omega_{\boldsymbol{k}}}{m}}, \tag{95}$$

$$\tilde{c}_{\boldsymbol{k}}^m = \frac{g \mu_B \tilde{B}_{\boldsymbol{k}}}{2}, \tag{96}$$

we can conveniently write the total Hamiltonian as

$$\hat{\mathcal{H}} = \hat{\mathcal{H}}_M' + \sum_\kappa \omega_{\boldsymbol{k}} \lambda_{\boldsymbol{k}} \hat{b}_\kappa^\dagger \hat{b}_\kappa - \sum_j \sum_\kappa \tilde{c}_{\boldsymbol{k}}^e \sqrt{\frac{1}{m\omega_{\boldsymbol{k}}}} \hat{\boldsymbol{p}}_j \left( \tilde{\boldsymbol{u}}_\kappa(\boldsymbol{R}_j) \hat{b}_\kappa + h.c. \right) \\ - \sum_j \sum_{\boldsymbol{k}} \tilde{c}_{\boldsymbol{k}}^m \hat{\boldsymbol{\sigma}}_j \left( \tilde{\boldsymbol{u}}_{\perp,\kappa}(\boldsymbol{R}_j) \hat{b}_\kappa + h.c. \right). \tag{97}$$

Defining the coupling operator

$$c_{\boldsymbol{k}} \hat{\mathcal{C}}_\kappa = \sum_j \left( \tilde{c}_{\boldsymbol{k}}^e \sqrt{\frac{1}{m\omega_{\boldsymbol{k}}}} \hat{\boldsymbol{p}}_j \tilde{\boldsymbol{u}}_\kappa(\hat{\boldsymbol{R}}_j) + \tilde{c}_{\boldsymbol{k}}^m \hat{\boldsymbol{\sigma}}_j \tilde{\boldsymbol{u}}_{\perp,\kappa}(\hat{\boldsymbol{R}}_j) \right), \tag{98}$$

we can finally write the Hamiltonian as

$$\hat{\mathcal{H}} = \hat{\mathcal{H}}_M' + \sum_\kappa \tilde{\omega}_{\boldsymbol{k}} \hat{b}_\kappa^\dagger \hat{b}_\kappa - \sum_\kappa c_{\boldsymbol{k}} \left( \hat{\mathcal{C}}_\kappa \hat{b}_\kappa + h.c. \right). \tag{99}$$

Which is formally identical to Hamiltonian (17), the only difference lies in how $\tilde{\omega}_{\boldsymbol{k}}, c_{\boldsymbol{k}}, \hat{\mathcal{C}}_\kappa$ and $\hat{b}_\kappa$ are defined in this case.

## B  Entirely Hamiltonian formulation

The effective Hamiltonian (41) derived in the previous subsection can also be deduced directly with an alternative derivation that does not rely on the path integral approach and the definition of an effective action. Instead, we borrow from Hepp and Lieb's original study of equilibrium superradiance [26] the following bounds for the partition function

$$\tilde{Z} \leq Z \leq e^{\beta \sum_\kappa \omega_{\boldsymbol{k}}} \tilde{Z}, \tag{100}$$

where $Z$, the proper partition function, and $\tilde{Z}$, its lower bound, are defined as

$$Z = \mathrm{Tr}_M \left( \int \prod_\kappa \frac{d^2 z_\kappa}{\pi} \langle \boldsymbol{z} | e^{-\beta \hat{\mathcal{H}}} | \boldsymbol{z} \rangle \right), \tag{101}$$

$$\tilde{Z} = \mathrm{Tr}_M \left( \int \prod_\kappa \frac{d^2 z_\kappa}{\pi} e^{-\beta \langle \boldsymbol{z} | \hat{\mathcal{H}} | \boldsymbol{z} \rangle} \right), \tag{102}$$

with $|\mathbf{z}\rangle = \bigotimes_\kappa |z_\kappa\rangle$ and

$$\langle \mathbf{z}|\hat{\mathcal{H}}|\mathbf{z}\rangle = \hat{\mathcal{H}}'_M + \sum_\kappa \tilde{\omega}_{\mathbf{k}}|z_\kappa|^2 + \sum_\kappa c_{\mathbf{k}}\left(\hat{\mathcal{C}}_\kappa z_\kappa + h.c.\right). \tag{103}$$

In the thermodynamic limit $N \to \infty$ and provided the number of modes does not scale with $N$, the correction term $\exp\left[\beta \sum_\kappa \omega_{\mathbf{k}}\right]$ in the upper bound becomes negligible and the partition function is given exactly by $\tilde{Z} = Z$. Then, we can define an effective Hamiltonian as

$$\tilde{Z} = \text{Tr}_M\left(\int \prod_\kappa \frac{d^2 z_\kappa}{\pi} e^{-\beta \langle \mathbf{z}|\hat{\mathcal{H}}|\mathbf{z}\rangle}\right) = \tilde{Z}_0 \,\text{Tr}_M\left(e^{-\beta\hat{\mathcal{H}}_{\text{eff}}}\right). \tag{104}$$

Note that in this context the $\{z_\kappa\}$ are complex numbers. We are thus interested in computing

$$\int \prod_\kappa \frac{d^2 z_\kappa}{\pi} e^{-\beta \langle \mathbf{z}|\hat{\mathcal{H}}|\mathbf{z}\rangle}, \tag{105}$$

which is just a collection of multidimensional complex Gaussian integrals. The resulting Hamiltonian is simply

$$\hat{\mathcal{H}}_{\text{eff}} = \hat{\mathcal{H}}'_M - \sum_\kappa \frac{c_{\mathbf{k}}^2}{\tilde{\omega}_{\mathbf{k}}}\hat{\mathcal{C}}_k\hat{\mathcal{C}}_k^\dagger, \tag{106}$$

just like Eq. 41. Note that in order to carry out the Gaussian integral, one has to be able to write

$$e^{-\beta\left(\hat{\mathcal{H}}'_M + \sum_\kappa \tilde{\omega}_{\mathbf{k}}|z_\kappa|^2 + \sum_\kappa c_{\mathbf{k}}(\hat{\mathcal{C}}_\kappa z_\kappa + h.c.)\right)} = e^{-\beta\hat{\mathcal{H}}_M}e^{-\beta\left(\sum_\kappa \tilde{\omega}_{\mathbf{k}}|z_\kappa|^2 + \sum_\kappa c_{\mathbf{k}}(\hat{\mathcal{C}}_\kappa z_\kappa + h.c.)\right)} \tag{107}$$

which holds because the matter Hamiltonian and the coupling operator commute $[\hat{\mathcal{H}}_M/N, \hat{\mathcal{C}}_\kappa/N] = 0$ in the thermodynamic limit $N \to \infty$. To see this, note that both $\hat{\mathcal{H}}_M$ and $\hat{\mathcal{C}}_\kappa$ are extensive and both can be written as a series in powers of the position and momentum operators, the commutation of position and momentum yields a dirac delta for the $i$th and $j$th particle, eliminating one of the two sums over the N particles. The result of the commutator is still extensive, so its division by $N^2$ will scale as $1/N$. The Gaussian integrals also yield $\tilde{Z}_0 = \prod_\kappa \tilde{Z}_\kappa$, where $\tilde{Z}_\kappa = 1/\beta\tilde{\omega}_{\mathbf{k}}$ is the high-temperature limit of the partition funtion of a Harmonic oscillator of frequency $\tilde{\omega}_{\mathbf{k}}$, i.e. $\tilde{Z}_\kappa = \lim_{T \to \infty} Z_\kappa$.

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
