# Peer review of "Effective theory for matter in non-perturbative cavity QED"

_SciPost Physics, doi:SciPost Phys. Lect. Notes 50 (2022)_

## Round 3 · Referee Report · Anonymous (Referee 1) · 2022-1-5

Strengths

The article is well written and well organized. The formalism is clear, and definitions, constants and variables are well organized. All mathematical passages are in general well explained.
A few previous results from other groups are well summarized in an homogeneous framework with unified notation.

Weaknesses

There is hardly anything new here. All the results are just a summary of previous results.

Report

My summary: the authors first introduce the most general Hamiltonian describing light-matter interactions in a non-relativistic framework, also including the Zeeman term to describe spin-magnetic field interactions. After a couple of manipulations and an approximation (eq. 13) they rewrite the light-matter Hamiltonian in a more compact form with a minimal set of parameters. After a short discussion on the usual long wavelength approximation they present the problem of finding the partition function of the system in the path integral formalism. They introduce the Matsubara frequencies and they integrate out the photonic degrees of freedom, ending up in a non-local action in imaginary time for the matter's degrees of freedom only. At this stage everything is still exact, except for the approximation explained in eq. 13. Then they invoke the thermodynamics limit, arguing about a rescaling 1/\sqrt{N} of the photonic operators. In this way the photonic commutation relations are negligible and one can make the effective matter-matter action local in imaginary time. From this effective action one can derive an effective Hamiltonian that includes matter degrees of freedom only. Once this step is clarified they make a short comment on how to compute observables from the partition function and they proceed to test the effective light-matter description on previous results. In particular they discuss a) the absence of so-called "photon condensation" in a single uniform mode cavity, and the presence of photon condensation in a multi-mode cavity including the Zeeman magnetic coupling. b) how to obtain an effective description for a free 2D electron gas in a cavity c) how to obtain an effective description for magnetic molecules in a cavity

In my opinion the article do not meets the acceptance criteria, mostly due to the absence of new results/perspectives. The information content is more suitable for lecture notes about basics usage of path integral in light-matter interactions rather than a scientific research article. If rewritten in terms of lecture notes the authors may think a re submission to Scipost lecture notes. Otherwise I would not recommend it for Scipost physics.

I will explain here the reasons of my negative recommendation.

Major reasons for rejection: 1) the authors put much emphasis on the derivation of this effective matter action which is only used to re-confirm results which are already well understood. It is surprising that they derive a so nice effective Hamiltonian, with claims of absolute generality and exactness, but then they do not use it to produce any new physical prediction/insight. 2) Derivation of effective actions/Hamiltonians from Feynman-Vernon influence functional to Born-Markov adiabatic elimination of fast variables is a very well known and standard procedure. In my opinion the mere derivation of an effective theory is definitely not enough if it is not used to pursuit new physical predictions which were not achievable before, or to highlight the differences between different methods and effective description, testing them on well known problems. Unfortunately the authors missed both points, making the content of this manuscript of little relevance.

Other concerns: 3) The authors stress a lot on the full generality of their effective Hamiltonian. I have some doubts on this point. Indeed the derivation of the effective action itself does not seem so general as they claim it is. First, it is not clear at all to me what are the consequences of the approximation in eq. 13, and why the authors do not explain it. Second, the thermodynamic limit seems to hide a perturbative limit. Indeed Eq.34 seems to me a mean-field assumption that turns out to be exact due to the thermodynamic limit (something which is rather standard in the literature of the Dicke model). It seems to me that this is a consequence of taking the cavity volume to infinity, which makes the light-matter coupling infinitesimal. The \sqrt{N} enhancement makes the coupling finite in the end, but only at the collective level. The single particle coupling to each mode is still infinitesimal. Indeed the effective Hamiltonian derived in this article could be easily derived by standard perturbation theory under the assumption that the photon energy scales are much faster than the matter ones and the single particle coupling to each mode is very small. I think that, in order to sustain their claims, the authors should provide comparison with other methods to derive such effective interaction, clarify all the assumptions of each methods and to highlight eventual discrepancies. 4) In certain parts of the manuscript I feel a lack of physical intuition. For instance, what is the difference between photon condensation and standard ferromagnetism? In Sec 5 A and C, it seems that the author just re-derived the Heisenberg model of spin-spin interaction. These mathematical passages without a clear physical interpretation in simple terms could rise confusion. 5) Taking the thermodynamic limit in 3D means to take an infinite cavity volume. This is basically the free space limit where the cavity do not make any difference and where boundary effects are completely negligible (image charges and currents). Do the authors just recover free space electromagnetism? 6) The authors stress on the exactness of their derivation which respect gauge invariance. However there is no trace of a proof or at least a convincing comment that this is true. I may believe that what they say is true, but I feel that those are subtle things that have to be taken with care. I have in mind the "Coleman-Weinberg mechanism" which involve a similar integration of the photon's field and the derivation of an effective action (Ref: https://doi.org/10.1103/PhysRevD.7.1888). Hiding in the assumptions of the calculation, gauge invariance was broken and this was recognized later (https://doi.org/10.1103/PhysRevD.9.1686), raising a dispute that went on for years. I cannot judge if here we have the same issues, but I wish to see more care from the authors when they make certain claims. 7) I think that the authors should also provide more information about next order corrections of the effective Hamiltonian. This might be important to understand gauge invariance (see above), but also to understand how this method is different/similar to other methods and perhaps useful to provide new physical predictions. 8) Are all the expressions convergent? I see that there are sums over "k" that seems to be UV divergent. I did not check explicitly, but did the authors do it? For instance, the effective Hamiltonian Eq. 49 seems to give a divergent self energy contribution in 3D. It seems to me a contribution similar to the mass renormalization term of the Lamb shift. If this is the case, how do they deal with it? 9) The authors expose their approach as "operational" and they keep arguing about experimental relevance. However in the manuscript there is no trace about a clear discussion on observables, what can be measure and, at least, some rough estimates on currently achievable experimental settings.

Minor comments: 10) First line below Eq. 1, I think there is a typo on the commutation relations.

  • validity: ok
  • significance: poor
  • originality: ok
  • clarity: ok
  • formatting: good
  • grammar: good

Author:  Juan Román-Roche  on 2022-03-09  [id 2277]

(in reply to Report 1 on 2022-01-05)

1) We respect the criticism, but we would like to argue that new “physical prediction/insight” were already provided in a previous publication where we applied a more barebones version of the theory to a system of magnetic molecules [10.1103/PhysRevLett.127.167201]. With the current manuscript, we wanted to take the opportunity to develop and highlight the theory in its general version and how it works in some models of current/recent interest. Moreover, following the referee’s comments (see Comment 2) we added a new section dedicated to a thorough testing of the validity of the method, including a comparison with other commonly used methods.

2) This is a fair criticism. In the new version of the manuscript we add a new section that tackles this issue. The validity of the effective theory is tested on the Dicke model by highlighting the differences with common techniques based on the adiabatic elimination of fast variables. We also address its applicability to finite size systems. We copy the conclusions here “These numerical results imply an important advantage of our effective theory in regards to its applicability to any light-matter coupling regime, hence the surname non-perturbative. They also prove that it is not equivalent to standard perturbation theory.”

3) When we refer to the Hamiltonian in Eq. (17) as general, we say it to mean that can describe a broad range of models, not that it describes all those models exactly. For instance, it is true that for a gas of charged particles the approximation of momentum conservation has to be applied. This restricts the validity of the Hamiltonian to gases where momentum is conserved. Nevertheless, the Hamiltonian can also describe a system of localized emitters, this time without approximations (See our Sec III B). Furthermore, these emitters can be electric of magnetic dipoles. The Hamiltonian in Eq. (17) also accommodates any cavity-geometry, not no mention that the bosonic subsystem need not originate from the microscopic description of a cavity, it may also describe a system of magnons for example. We believe that some emphasis on generality is justified.

Regarding the particular approximation in Eq. (13), we concede in the text that our analysis of its validity is not complete. But we agree with the referee that some additional clarification on its consequences is required. We have updated the manuscript accordingly. We copy here the new paragraph: “The approximation is exact for a homogeneous gas. Furthermore, for gasses that have a second order phase transition to a non-homogeneous phase, the approximation is valid in the vicinity of the transition [10.1103/PhysRevB.100.121109]. This suggests that the resulting model can be used to predict and locate transitions to non-homogeneous gas phases. It is expected that the effect of the approximation will be quantitative and will skew the calculation of observables in the non-homogeneous phase.”

Finally, as we detailed in our response to comment 2), in the new section of the manuscript we compare the effective theory against standard perturbation theory. They are shown not to be equivalent in all regimes. Although they do coincide in the fast-cavity limit, as the referee rightfully points out.

4) For a magnetic material, in the context of Sec. 5C, standard ferromagnetism arises from the exchange interaction (Pauli’s exclusion principle) whereas photon condensation, or rather its manifestation on the matter subsystem in the form of a cavity-mediated position-dependent all-to-all spin-spin interaction, arises due to the coupling to a sufficiently confined EM field. They are unrelated in their origin. Furthermore, the exchange interaction depends on the overlap of the electron’s wave functions, and thus is rather short-ranged. In contrast, the cavity mediated interaction is long-ranged, for instance for two spins placed at antinodes of the cavity field. For a ferromagnetic material coupled to a cavity, the intrinsic and cavity-mediated magnetic interactions are combined constructively or destructively (depending on the geometry of the coupling to the cavity and the nature of the intrinsic interaction). Now, as we show in Sec. IV C, matter and light observables are related, so it is the case that if a magnetic material coupled to a cavity orders ferromagnetically (be it due to intrinsic or mediated interactions, or their combination) a photonic population appears in the cavity. It is in this sense that photon condensation and ferromagnetism can be linked.

A similar reasoning can be applied to a ferroelectric material, with the caveat that electric coupling is affected by the no-go theorem for homogeneous coupling to the cavity fields.

5) We want to clarify that even in 3D the thermodynamic limit is not trivially free space electromagnetism. This is because the number of particles scales accordingly, filling this infinite space. Intuitively, when we neglect the effect of electromagnetic field on a material in free space, it is because the material fills a negligible volume of this free space. Because the energy of the EM vacuum is spread across all of free space, only a negligible amount couples to the material. When one considers a cavity, it has the effect of confining the EM field such that most of it interacts with the material. Any cavity-mediated effects originate from this confinement. Making the cavity volume infinite does not negate this effect if the material grows accordingly and the filling factor is maintained. Looking at the Hamiltonian, it can be seen that the coupling term is extensive and does not become negligible in this limit. This is the mathematical manifestation of the above argument.

6) The referee is indeed right in their concern about gauge dependencies of the effective theory. It is, in a sense, gauge dependent. However, this does not mean that it cannot be used to make gauge invariant predictions nor does it conflict with our emphasis about gauge invariance during the derivation of the microscopic model. The reason for our insistence on gauge-invariance during the Introduction and the derivation of the microscopic models, up to the Hamiltonian in Eq. (17), stems from the fact that it has been a historically controverted topic in the field. The omission of the $A^2$ term leads to the incorrect prediction of superradiance in models that lack it. The truncation of matter energy levels is also gauge-sensitive. Our emphasis on working with a gauge-invariant microscopic model aims to avoid these pitfalls. Now, because our effective theory traces out the “light” degrees of freedom, it is intrinsically gauge dependent. This is because the separation between “light” and “matter” subsystems is itself gauge dependent. Photons in the Coulomb gauge are excitations of the transverse cavity fields, whereas in the dipole gauge they also incorporate longitudinal fields associated with the polarization field in the matter. This important issue is not downplayed in the manuscript, we devote the third paragraph of the introduction to discuss it. Nazir and Stokes have discussed this concept extensively, see arXiv:2009.10662. Our choice of the Coulomb gauge responds to this issue. In the Coulomb gauge the EM fields are transverse and thus electromagnetic interactions within the material appear as instantaneous Coulomb interactions included in the matter subsystem. This makes sense in this context because these interactions are fully relevant when the material is in free space, in fact, they help explain much of the structure and properties of most materials. By choosing the Coulomb gauge, the effective interactions that appear on the effective action are solely mediated by the transverse fields of the cavity and the intensity is dependent on the degree of confinement of the EM fields. I.e. they reveal the impact of coupling the material to a confined EM field, as opposed to it being in free space. Any observable, consider for instance the full electric field or some correlation function, has its expression in the Coulomb gauge, which may include a combination of light and matter operators. The expectation values of the former can be related to the expectation values of other matter operators using the recipes in Sec. IV C. The expectation value of the resulting matter-only operator can be computed within the effective theory. The key to making gauge-invariant predictions on observables is thus to express them in terms of light and matter operators as prescribed by the Coulomb gauge, as it is in this gauge where the light is traced out.

7) This issue is also addressed in the new section of the manuscript. As we mention there, “our effective Hamiltonian was derived in the $N \to \infty$ limit. As such, it can be regarded as a zeroth order perturbation theory with $1/N$ as the perturbative parameter. The extension to a full-fledged perturbation theory that can be taken to any order in $1/N$ is a work in progress.”. To alleviate the lack of an analytical approach to the N →∞ limit, we perform a numerical study of the finite size scaling of the theory. As we already commented above, we also show that perturbations in $1/N$ are not equivalent to standard perturbation theory on the light matter coupling. Finally, as per our response to comment 6), we believe this is unrelated to gauge issues.

8) The referee is correct in pointing out that these sums over k in the effective theory can lead to UV divergences. This is pragmatically resolved by introducing a frequency cutoff. This is often the case in cavity QED, and it is physically justified by the limitations of the hardware, i.e. real cavities cannot host infinite frequency modes. In the particular case of the Hamiltonians in Eq. 49 or Eq. 76, it does indeed lead to a mass renomarlization. For the case of a 2D electron gas in the long wavelength limit this model and its mass renormalisation is discussed in Ref. 10.1103/PhysRevResearch.4.013012. The cavity field is shown to dress the electrons, increasing their effective mass. Ultimately, the cutoff cannot be arbitrarily large: the renormalised mass is shown to become infinite when the cutoff is raised to the Landau pole, signalling the breakdown of the effective theory.

UV cutoffs are also commonly employed in dissipative models with ohmic spectral functions. In the context of the path integral, what we denote as the kernel $\bar{\bar K}_\kappa(\tau)$ in Eq. (39) is divergent in models with ohmic damping. This divergence is again tamed by introducing a high frequency Drude cutoff to the spectral function [10.1007/3-540-45855-7_1].

9) We have added a paragraph on Sec. IV C (Relation between matter and light observables) to discuss the probing of cavity qed materials and how the observables are related to the relations that we present in this section. We copy it here for convenience:

“These relations may be important for computing measurable outputs of a cavity QED material. For example, if we probe the cavity-matter system through the transmissivity of the former, the transmitted signal is proportional to the dynamical response of the system. This is in turn related to two point correlators of bosonic fields that can be directly computed from relation (46) or analogues”

---

## Round 3 · Referee Report · Anonymous (Referee 2) · 2022-1-13

Strengths

1-The manuscript is written clearly and provides a good overview over the challenges encountered when dealing with light-matter systems in the ultrastrong coupling regime.
2-The derivation of the effective theory is worked out in detail and explained in a very pedagogical manner.

Weaknesses

1-The general procedure resulting in the effective Hamiltonian (integrating out certain degrees of freedom in a coherent state path integral approach) is not new and was previously applied in various other settings.
2-Other than the derivation of the effective Hamiltonian the manuscript does not contain any new results. Only already known results are reproduced.
3-The advantages or disadvantages of the presented approach and it's comparison to alternative perturbative approaches are not covered.

Report

In this manuscript the authors present a detailed derivation of an effective model for matter in cavity QED in the ultrastrong coupling regime. The obtained effective model is then applied to three special cases and well known results are reproduced. The main "result" of the manuscript is a model derivation, which in most parts follows the standard approach of integrating out certain degrees of freedom in a path-integral formulation. While this derivation presented in chapters III and IV is presented in a nice and pedagogical manner, the rest of the manuscript does not contain any new results which go beyond what was already obtained via alternative models.

Therefore, I don't think the manuscript meets the acceptance criteria of 'SciPost Physics' and I cannot recommend the publication of the manuscript in its current form. However, I think the manuscript can be a valuable pedagogical piece for 'SciPost Physics Lecture Notes'.

More detailed comments:
1-In the caption of Figure 1 the authors claim that the Hamiltonian presented in panel b) describes a broad range of models. I am not so sure about this statement. When deriving this Hamiltonian in section III A, the authors start with a very general Hamiltonian in Eq. (1). However, when going from Eq. (12) to Eq. (13) the authors perform a quite drastic assumption by leaving only momentum-conserving terms. I think this assumption restricts the model to a much smaller class of systems where the Hamiltonian in Eq. (17) is actually applicable.

2-The authors emphasize that the presented theory is non-perturbative. In fact, they never explain in detail what non-perturbative means in the context of the presented model.

Minor comment:
In the third line of page 4 there is a duplicate 'of'.
  • validity: ok
  • significance: low
  • originality: ok
  • clarity: good
  • formatting: good
  • grammar: good

Author:  Juan Román-Roche  on 2022-03-09  [id 2276]

(in reply to Report 2 on 2022-01-13)
Category:
answer to question
objection
reply to objection
validation or rederivation

1) First, our claim that “the Hamiltonian presented in panel b) describes a broad range of models” may seem pretentious at face value, but it simply refers to the fact that it is a general Hamiltonian in which some bosonic modes are linearly coupled to an unspecified subsystem. It can be considered general for two reasons, because the nature and geometry of the coupling is not fixed and because many microscopic models, either exactly or approximately, can be cast into this form. Naturally, if approximations are involved in reaching the Hamiltonian in Eq. (17), this restricts its applicability to the regime where the approximations are valid. But this consideration is specific to particular models. This is related to the second part of the comment, which rightfully points out that in reaching this Hamiltonian from the microscopic model of a gas of charged particles coupled to a cavity, we made an assumption about the conservation of momentum. However, this approximation is specific to this model. Other models do not require it and still map exactly to the Hamiltonian in Eq. (17), for instance a system of localized emitters, which is also discussed in the main text (although the calculations are in the Appendix A). Another example of a system that exactly maps to the Hamiltonian in Eq. (17) is a system of magnetic spins, such as the one discussed in Sec. VI C. Surely there are other examples outside of the realm of cavity QED.

In summary, we consider the claim to be fair. We do not claim the Hamiltonian in Eq. (17) to be all-encompassing, that would be an exaggeration. But “broad” seems adequate for a Hamiltonian that can describe bosonic, fermonic and spin systems coupled to bosonic modes with an arbitrary geometry of the coupling.

2) This criticism is addressed in the new section of the manuscript, were the theory is compared with other effective theories and the surname non-perturbative is contextualised. We copy here the relevant paragraphs:

“In this regard, it must be noted that our effective Hamiltonian was derived in the $N \to \infty$ limit. As such, it can be regarded as a zeroth order perturbation theory with $1/N$ as the perturbative parameter. The extension to a full-fledged perturbation theory that can be taken to any order in $1/N$ is a work in progress. However, in this context we will use the term perturbative to refer only to parameters that affect the light-matter coupling regime, such as the light-matter coupling strength or the detuning between the cavity frequency and the characteristic energy scale of the matter subsytem. It is in this sense that our theory is non-perturbative. The simplest example of a perturbative theory would be standard perturbation theory on the light-matter coupling. To yield an effective Hamiltonian with perturbation theory, a separation of energy scales (large detuning) is required, with the more energetic sector of the spectrum mediating effective interactions on a low-energy subspace. […] These numerical results imply an important advantage of our effective theory in regards to its applicability to any light-matter coupling regime, hence the surname non-perturbative. They also prove that it is not equivalent to standard perturbation theory.”

---

## Round 4 · Referee Report · Anonymous (Referee 1) · 2022-3-30

Report

I appreciate the detailed answers provided by the authors, however they were not able to change my initial recommendation. I thus re-affirm that this article does not fully meet the criteria to be published in SciPost Physics. Perhaps I can still suggest it for SciPost Lecture notes, since it covers in quite details the derivation of effective Light-Matter interactions and it reports several existing examples.

My rejection recommendation is mainly motivated by the fact that the authors provided only minor changes to the manuscript, while it should be drastically changed to fulfill the acceptance criteria.

Here I want to address the answers provided by the authors poit by point: 1) If all the results in this article were already known then it is more a review or a lecture note rather than an article containing new original results. Moreover the new paragraph about the Dicke model provided by the authors is a quite trivial example. It doesn't bring that much extra information/understanding and instead it makes the article longer and less fluid to read. 2) Basically the same comment as in 1) 3) The authors didn't really resolve my doubts and the role of their assumptions is still unclear. 4) What I meant in my comment was: if I have a magnet and I put it in a cavity (and nothing happens) do I have a photon condensate? Because in that case, due to the magnetostatic field the expectation value of "a" is different from zero in equilibrium. Then I'd say that using the term "photon condensation" in this context is rather misleading. 5) Here I meant that if I think the cavity as the boundary of the system and I take the cavity volume to infinity the effect of the boundary should be quite negligible, even if the cavity is fully filled. Then electromagnetism still plays a role but it is not anymore "cavity QED" in the sense that mode quantization doesn't play any role. 6) I kind agree with the authors. However it seems to me that what they say is simply "do the calculation correctly without mistakes". Still it is not clear whether all their assumptions and approximations respect gauge invariance. 7) The Dicke model is quite easy and perhaps trivial example that could hide features of the author's effective treatment which are critical in other less trivial systems. I do not think that their example gives really any useful information. I think I expected a deeper analysis on the general model rather than a single example with one of the simplest toy model on the market. 8) I agree with the authors that one introduce cut-offs to cure divergences. It would be interesting to see a proper calculation on this side and to check which kind of physical predictions may arise from there, for instance regarding the Lamb-shift. 9) The authors added a vague sentence about observation. I do not think it adds nothing more than was already written and thus it doesn't solve my doubt.

---

## Round 4 · Referee Report · Anonymous (Referee 2) · 2022-3-31

Report

I appreciate the time and effort the authors took to improve the manuscript and answer my initial report. The authors responded to my initial criticism mainly by adding a section in the revised version of the manuscript (section V) where they compare the effective theory presented in this work with other known effective theories. To this end they use the paradigmatic example of the Dicke model and compare their effective Hamiltonian with other effective Hamiltonians obtained via i) a Polaron transformation and ii) a Schrieffer-Wolff transformation. This novel section is a valuable addition to the manuscript and it also clarifies my original question why the presented theory can be called "non-perturbative".

While the new section increased the quality of the manuscript, it still contributes mainly to the pedagogical aspect of the manuscript. As I already pointed out in my initial report the "result" presented in this work is the model derivation. I still think that applying the introduced methodology to a particular problem, which cannot be understood by other (e.g. perturbative) effective theories would be necessary to fulfill the criteria for publication in SciPost Physics.

Therefore, I stay with my original assessment of this manuscript and I think it is not suitable to be published in SciPost Physics.

---

## Round 4 · Author Response

Dear Editor,

I am resubmitting a revised version of the manuscript “Effective theory for matter in non-perturbative cavity QED”. In this revised manuscript, we have made changes to address the referees’ comments and suggestions, which have resulted in the addition of a new section to the paper, as well as more minor changes.

The referees criticised the lack of novelty in the previous version of the manuscript, which resulted in your editorial recommendation for SciPost Physics Lecture Notes. We respect the decision, and would feel honoured to be featured in SciPost Physics Lecture Notes, as we acknowledge the pedagogical caliber of its publications. However, we find that appearing as a Lecture Note would mislead potential readers of the manuscript into thinking that it contains nothing new, that it just reviews well-known concepts. While the idea of an effective action dates back to the works of Feynman and Vernon on the influence functional, its application to the field of cavity qed materials, ending up with an exact effective Hamiltonian valid for all light-matter-coupling regimes, is definitely novel. Moreover, the new section is dedicated to testing the validity of the theory, highlighting its differences against other effective theories, which is something that the referees pointed out the previous version lacked. With this concern in mind, we would like to ask you and the referees to reconsider your recommendation for SciPost Physics Lecture Notes.

Detailed responses to each referee report are provided separately via the “Reply to the above Report” functionality on the website.
An overview of all the changes in the updated manuscript is provided below.

Sincerely,
Juan Román-Roche

---

## Round 4 · List of Changes

• In the first line below Eq. 1, corrected typo on the commutation relations
  • In the third line of page 4 corrected duplicate 'of'.
  • Added new section “TESTING THE VALIDITY OF THE EFFECTIVE THEORY”
  • Minor changes to the Abstract, Introduction, Summary of main results and Conclusion to accommodate the new section.
  • Minor change to Section III A to clarify the consequences of the approximation made in Eq. (13)
  • Minor change to Section VII to include citation.
  • Added paragraph to Section IV C following Comment 9 of Referee 1.

---

## Editorial Decision

published